# Improving Native CNN Robustness with Filter Frequency Regularization

**Jovita Lukasik**[*]                                                     *jovita.lukasik@uni-siegen.de*
*University of Siegen*

**Paul Gavrikov**[*]                                                      *paul.gavrikov@hs-offenburg.de*
*IMLA, Offenburg University*

**Janis Keuper**                                                          *janis.keuper@hs-offenburg.de*
*IMLA, Offenburg University*

**Margret Keuper**                                                        *keuper@mpi-inf.mpg.de*
*University of Mannheim*
*Max Planck Institute for Informatics, Saarland Informatics Campus*

**Reviewed on OpenReview:** *https://openreview.net/forum?id=2wecNCpZ7Y*

## Abstract

Neural networks tend to overfit the training distribution and perform poorly on out-of-distribution data. A conceptually simple solution lies in adversarial training, which introduces worst-case perturbations into the training data and thus improves model generalization to some extent. However, it is only one ingredient towards generally more robust models and requires knowledge about the potential attacks or inference time data corruptions during model training. This paper focuses on the native robustness of models that can learn robust behavior directly from conventional training data without out-of-distribution examples. To this end, we study the frequencies in learned convolution filters. Clean-trained models often prioritize high-frequency information, whereas adversarial training enforces models to shift the focus to low-frequency details during training. By mimicking this behavior through frequency regularization in learned convolution weights, we achieve improved native robustness to adversarial attacks, common corruptions, and other out-of-distribution tests. Additionally, this method leads to more favorable shifts in decision-making towards low-frequency information, such as shapes, which inherently aligns more closely with human vision.

## 1 Introduction

Modern convolutional neural networks (CNNs) (He et al., 2016; Liu et al., 2022; Tan & Le, 2019) show a steady increase in performance in terms of test accuracy on a wide range of learning tasks. Yet, most models suffer from a low generalization ability, even when faced with small domain shifts. To improve the low generalization ability, previous work focused on aspects such as aliasing (Zhang, 2019; Zou et al., 2020; Li et al., 2020; Grabinski et al., 2022b;a), the padding operations (Gavrikov & Keuper, 2023), the training schedule (Lopes et al., 2019; Saikia et al., 2021), or analyzing the image feature spectrum (Geirhos et al., 2019; Wang et al., 2020). In addition, introducing perturbed images into the training data, known as adversarial training (AT) (Madry et al., 2018), can alleviate low generalization to some extent. However, AT is not the cure-all to improve network robustness and tends to overfit on training attacks (Tramèr & Boneh, 2019; Rice et al., 2020; Yu et al., 2022). Intuitively, the adversarial attack used during training becomes an in-domain sample of the model, while its robustness to new out-of-domain samples (*e.g.* a different adversarial

---

[*]Equal contribution.

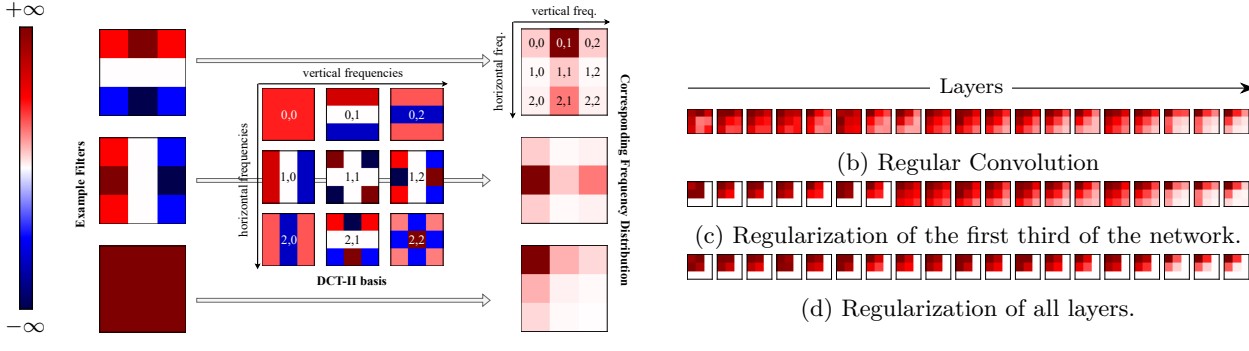

(a) Computation of frequency distributions

Figure 1: Our proposed regularization decreases the reliance on high-frequency information of a ResNet-20 trained on CIFAR-10 (b-d) as visible in mean DCT-II coefficients magnitudes (= frequency distribution; computation shown in (a)).

attack) is hard to anticipate. Saikia et al. (2021) show that AT can even increase the mean corruption error on ImageNet-C (Hendrycks & Dietterich, 2019). Therefore, we argue that AT can only be one ingredient towards building more robust models, while the main focus should rather be to encourage behavior that we call *native robustness*. We expect from natively robust models that they can learn robust behavior directly from the conventional training data. Thereby, robust behavior includes, on the one hand, a certain degree of adversarial robustness without being confronted with adversarial attacks during training, *i.e.* the model should not easily be fooled using attacks with very small perturbation budgets. Similarly, they should be robust against other perturbations such as common corruptions (Hendrycks & Dietterich, 2019) as long as corruption severities are low. On the other hand, robust behavior implies a better alignment with human perception, *i.e.* models should decide for a specific class more by the shape of an object than by its texture (Geirhos et al., 2019). Note that the expected degree of *specific* robustness can not be compared to the one obtained by techniques that specifically optimize for them, such as adversarial training. For instance, adversarial samples remain out-of-domain samples for such natively robust models. Yet, additionally training these natively robust models with AT should be complementary and have a further beneficial effect.

In this paper, we propose a new perspective on improving native robustness by investigating the frequencies in the learned network filters directly. Specifically, we propose to project CNN convolution filter weights into the frequency domain by applying a discrete cosine transformation (DCT-II). Although the resulting formulation is in principle equivalent to the commonly adopted CNN formulation, it provides direct access to the learned filter frequencies. Thereby, we aim to investigate the following research questions: (i) Which filter frequencies are predominantly learned in the layers of CNNs? (ii) Can we regularize the frequencies during the training process such as to increase the native robustness of the learned model?

We investigate these questions in the context of image classification - yet our approach bears the potential to be expanded to other tasks such as object detection and segmentation. First, we analyze the learned filter frequencies of modern CNNs and observe that they tend to have a low-frequency bias in deep layers, while filters of earlier layers of the network are either uniformly distributed in frequency space or even biased towards higher frequencies. In the latter cases, the convolution thus relies on high-frequency information. Contrary, adversarial training appears to shift the focus to low filter frequencies in early layers. To leverage this behavior, we introduce a regularization scheme, which increases the bias to low-frequencies in these early layers (see Figure 1 for a visualization). We evaluate the proposed spectral decomposition and regularization on different CNNs under distribution shifts in test data. Results on CIFAR-10, CIFAR-100 (Krizhevsky, 2009), SVHN (Netzer et al., 2011), MNIST (LeCun et al., 2010), Tiny-ImageNet (Le & Yang, 2015) and ImageNet (Deng et al., 2009) show increased native robustness[1]. In summary, we make the following contributions:

---

[1]Code: https://github.com/jovitalukasik/filter_freq_reg

- We observe that adversarial training results in a shift towards a low-frequency bias in the filter weights of early layers during the early phases of training (Section 3).
- Based on this observation, we propose a high-frequency penalization term in the weight space of convolution layers (Section 4) to mitigate the reliance on high-frequency information.
- Networks trained with this regularization become gradually, yet consistently, more robust against a wide array of out-of-distribution generalization tasks without reliance on AT or additional data - *i.e.* networks increase their native robustness (Section 5). Additional AT is complementary and further improves the measurable adversarial robustness to a variety of attacks.

## 2 Related work

**Robustness.** While modern neural networks yield accuracies close to or even beyond human performance, they seem to struggle with generalization to out-of-distribution data. In the context of adversarial attacks, it has been shown that minor, for the human eye barely perceivable perturbations can cause models to make wrong predictions with high confidence. Formally, let $f$ be a model parameterized by $\theta$, $\boldsymbol{x}$ an input sample with the corresponding class label $\boldsymbol{y}$, and $\mathcal{L}$ the loss function. Then adversarial attacks will attempt to maximize the loss $\mathcal{L}$ by finding an additive perturbation to an input sample $\boldsymbol{x}'$. To constrain their intensity, perturbations are sought in an $\mathcal{B}_\epsilon(\boldsymbol{x})$ ball centered at $\boldsymbol{x}$ with a radius of $\epsilon$.

$$\max_{\boldsymbol{x}' \in \mathcal{B}_\epsilon(\boldsymbol{x})} \mathcal{L}\left(f\left(\boldsymbol{x}';\theta\right),\boldsymbol{y}\right), \quad \mathcal{B}_\epsilon(\boldsymbol{x}) = \{\boldsymbol{x}' : \|\boldsymbol{x} - \boldsymbol{x}'\|_p \leq \epsilon\}. \tag{1}$$

with $\|\cdot\|_p$ depicting the $L^p$-norm. The most successful adversarial attacks are white-box attacks, where the attacker has full access to the attacked model. Often, these methods rely on gradient information, such as *projected gradient descent* (PGD) (Kurakin et al., 2017) where the attacker follows the gradient that maximizes the loss and then projects the perturbations back to $\mathcal{B}_\epsilon(\boldsymbol{x})$. Since PGD is computationally expensive, a faster, yet less successful attack that approximates the perturbations by the gradient sign and only performs one step has been proposed: *fast gradient sign method* (FGSM) (Goodfellow et al., 2015).

Regarding defenses in general, the most successful approach to tackle out-of-distribution shifts is *adversarial training* (Madry et al., 2018) where worst-case perturbations are reintroduced to the training data. Often, these methods are accompanied by additional external data (Carmon et al., 2022). Evaluating the defenses on a single attack can be misleading, due to the possibility of attack overfitting (Rice et al., 2020). Towards more reliable benchmarks, *AutoAttack* (AA) (Croce & Hein, 2020a) proposes an ensemble of various white- and black-box attacks such as *APGD* (Croce & Hein, 2020a), *FAB* (Croce & Hein, 2020b), and *Square* (Andriushchenko et al., 2020) and establishes the public *RobustBench* leaderboard. Benchmarks are constrained to $p = 2, \epsilon = 0.5$ on CIFAR-10, $p = \infty, \epsilon = 8/255$ on CIFAR-10/100, and $p = \infty, \epsilon = 4/255$ on ImageNet, respectively. However, these large thresholds are disputed as they generate easily detectable perturbations (Lorenz et al., 2022).

Unfortunately, the possibility of adversarial attacks is only a symptom of larger generalization issues. For example, neural networks fail to generalize under various corruptions such as weather conditions, changes in lighting, noise, and blurring (Dodge & Karam, 2017; Hendrycks & Dietterich, 2019). For fast and comparable benchmarks, common corruption datasets CIFAR-10-C, CIFAR-100-C, and ImageNet-C have been proposed (Hendrycks & Dietterich, 2019), which include 15 (+4 extra) types of corruption at increasing severity level (from 1 to 5).

Additionally, Geirhos et al. (2019) observed that CNNs are biased towards detecting textures of an image instead of the shape (cue-conflict), which is in contrast to human vision behavior that focuses on shape information, *i.e.* shape bias. To overcome this texture bias, they train on a stylized version of ImageNet to increase the shape bias of CNNs. For fast evaluation of out-of-distribution (OOD) generalization Geirhos et al. (2021) proposed a benchmark including 17 OOD datasets, from which 12 contain image perturbations and the other 5 are single manipulations of ImageNet (Deng et al., 2009): cue-conflicted texture vs. shape data, sketches (Wang et al., 2019), stylized images, edges, and silhouettes. They evaluate and compare more than 50 different networks to human performance to determine the gap between human and machine vision.

**Frequencies and robustness.** Recent work demonstrated the importance of learned frequencies for network robustness. Wang et al. (2020) demonstrated that CNNs initially rely on low-frequency information for prediction, but shift towards high-frequency information as training progresses. On the other hand, AT models predominantly classify based on low-frequency information. As texture information typically resides in higher frequency bands, this is a suitable explanation for the observations by Geirhos et al. (2019). As such, there is also a correlation between AT and a reduced texture bias (Geirhos et al., 2021; Gavrikov et al., 2023). Duan et al. (2021) exploit these findings by proposing an adversarial attack that drops DCT coefficients corresponding to high frequencies from inputs to fool neural networks. Yet despite the common assumption, adversarial attacks are not always targeting high-frequencies and the behavior depends on the dataset (Maiya et al., 2021; Abello et al., 2021; Bernhard et al., 2021; Ortiz-Jiménez et al., 2020).

Multiple works explore the desensitization of neural networks to HF from various angles to avoid AT: Lopes et al. (2019) randomly add noise to image patches, Saikia et al. (2021) regularize the feature maps produced by convolution layers in a dedicated two-stream architecture, and Grabinski et al. (2022a) introduce a downsampling approach within the frequency domain that removes aliasing-related high-frequency information. Huang et al. (2023) propose a frequency regularization in the context of adversarial training, which - contrary to the previous works - amplifies high frequencies, resulting in higher robust accuracy, at an impairment of clean accuracy. In contrast, we regularize high-frequency information directly in convolution filters to improve the native robustness and OOD generalization of the model, which is not limited to effects on adversarial training, but indeed can also improve the robust accuracy in the context of adversarial training, and eventually mitigate robust overfitting.

**Feature map vs. weight regularization.** We have previously outlined the role of HF in generalization within the existing literature. In this paragraph, we aim to emphasize the distinction between our proposed weight regularization method and prior research.
*Regularization vs. bandpass filtering:* Previous methods often lowpass-filter signals which leads to a hard smoothing of the resulting feature maps and the active deletion of information that may be necessary to predictions - especially in fine-grained classification problems. Instead, we only regularize the attenuation and thus effectively force the network to reweigh information without having to discard information.
*Data-independent and explicit attenuation:* By regularizing weights, we induce an explicit causal bias in the operator. Alternatively, an attenuation of feature maps would be implicit and would highly depend on the frequency distribution of the inputs. Additionally, attenuating the filters results in a local suppression (i.e., in the patch) of HF, while a (global) feature-map regularization would affect the entire scene.

**Basis decomposition.** The decomposition of convolution filters is typically studied in the context of compression, see Yaroslavsky (2014) for an overview. The majority of decomposition approaches convert the convolution layer weights to the frequency domain *e.g.* by utilizing the DCT-II-basis (Chen, 2004; Chen et al., 2016; Lo & Hang, 2019; Cheinski & Wawrzynski, 2020; Chen et al., 2022; Ulicny et al., 2022) to prune and compress the number of frequency components. But works also exist that transform the input images directly for better performance and generalization (Xu et al., 2020; Hossain et al., 2019). In detail, the *discrete cosine transform* (DCT) (Ahmed et al., 1974) maps an input signal into a frequency domain represented by cosine basis functions. In particular, the common DCT-II variant is used in JPEG compression, where it successfully compresses natural images (Wallace, 1992). These works mainly explore the fact, that data of multiple domains is not uniformly distributed in the frequency domain and is typically biased towards low frequencies (Singh & Theunissen, 2004; Ruderman, 1994). Gavrikov & Keuper (2022a;b) showed that the basis of convolution filter kernels obtained via SVD is often highly similar and independent of the architecture, learned task, or dataset. The identified bases have a striking similarity to the DCT-II basis.

Our realization of the DCT-II basis is similar to Ulicny et al. (2022) and other previous work, however, instead of compression, we explore an orthogonal direction and study the role of individual frequencies in training and apply regularization in the frequency space to improve generalization. DCT merely serves as a tool in our study.

## 3 Frequency analysis

In this initial analysis, we transform learned convolution filters to the frequency domain. We implement this by changing the basis of convolution weights to DCT-II, revealing the coefficients and therefore frequency information. Formally we define this as follows. Let $\mathbf{V}$ denote the $k \times k$-DCT-II basis. Then every basis vector $\mathbf{V}_{i,j}$ with horizontal frequency $j$ and vertical frequency $i$ is defined as:

$$\mathbf{V}_{i,j,m,n} = \cos\left[\frac{\pi i}{k}\left(m + \frac{1}{2}\right)\right]\cos\left[\frac{\pi j}{k}\left(n + \frac{1}{2}\right)\right]. \tag{2}$$

Every basis vector is additionally normalized to its $L^1$ length: $\mathbf{V}_{i,j} = \mathbf{V}_{i,j}/\|\mathbf{V}_{i,j}\|_1$. We show the DCT-II basis vectors for different kernel sizes $k$ in Figure 2. In principle, DCT-II could be replaced by any other

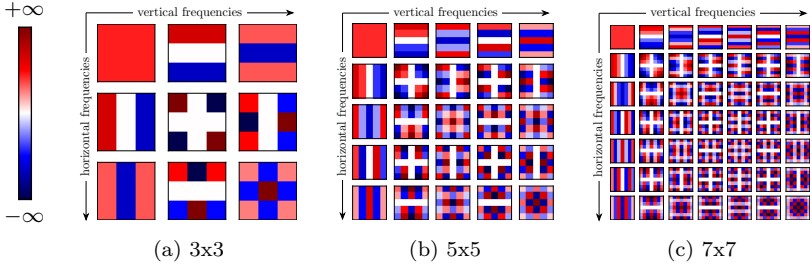

(a) 3x3     (b) 5x5     (c) 7x7

Figure 2: The full DCT-II basis for different resolutions.

frequency base such as a discrete Fourier or sine transform. Following the basis change, we visualize the average magnitude of coefficients in every convolution layer by heat maps (as shown in Figure 1). Having the frequency information at hand, we can directly analyze its distribution in common CNNs.

### 3.1 Analyzing learned convolution weights

We start by analyzing two modern networks trained on ImageNet without any robustness optimization techniques: EfficientNet-B0 (Tan & Le, 2019) and ConvNeXt-Tiny (Liu et al., 2022) (Figure 3). Our visualizations show that these CNNs do not always learn a uniform frequency spectrum utilization throughout the network. Earlier layers show a more uniform distribution of magnitude or are biased towards higher frequencies. However, deeper convolution layers instead reveal a salient bias towards low frequencies. Some layers even appear to discard a majority of high-frequency information.

In addition, we are interested in how adversarial training affects the frequency utilization in convolution filters. As shown from various angles (Wang et al., 2020; Geirhos et al., 2019; Saikia et al., 2021) robust models shift their bias to low-frequencies, as this reduces the possibility of overfitting on high-frequencies and therefore provides better generalization abilities. Thus, we expect that these results transfer to the frequency utilization in weight space to some extent. Indeed, Wang et al. (2020) stated that the very first

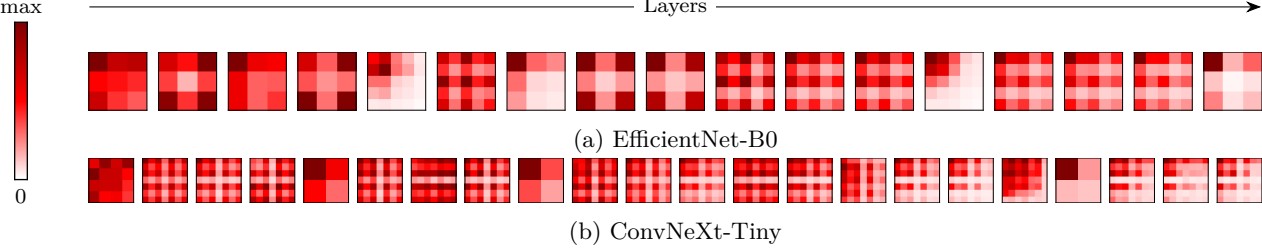

(a) EfficientNet-B0

(b) ConvNeXt-Tiny

Figure 3: Frequency distribution of each layer for trained (a) EfficientNet-B0 (containing $3 \times 3$ and $5 \times 5$ kernels) and (b) ConvNeXt-Tiny (containing $2 \times 2$ [downsampling layers], $5 \times 5$ [stem] and $7 \times 7$ kernels).

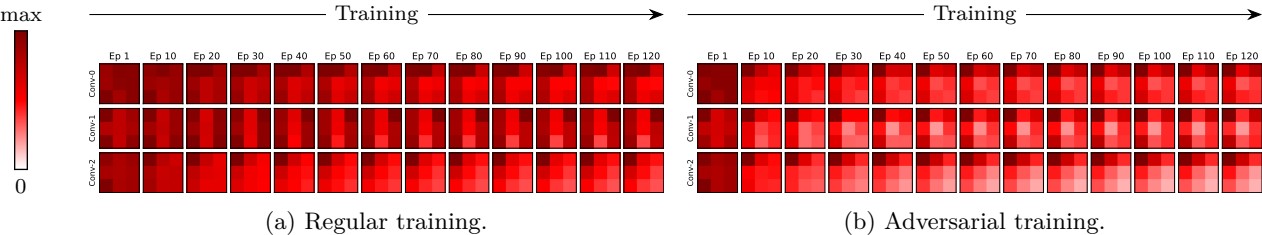

Figure 4: Evolution of the frequency distributions in the first three convolution layers of an EfficientNet-B0 in comparison between (a) regular and (b) adversarial training with CIFAR-10. Evolution plots for all layers and other architectures can be found in Appendix L.

convolution layer of AT CNNs learns smoother filters which equals to filters that are less reliant on high-frequency information than the equivalents in normally trained models. However, their frequency analysis was limited to the first initial layer, while we aim to provide a holistic analysis over the entire network. This is also backed by our previous observations showing that frequency utilization varies by depth. Further, their results do not appear to be representative of modern models that are trained under $L^\infty$-norm. Such models predominantly learn thresholding filters (Madry et al., 2018) independent of architecture and dataset (Gavrikov & Keuper, 2022b) that do not resemble "common" first layers as shown by Yosinski et al. (2014). As such, they are hardly smooth.

Exemplarily, we proceed by comparing an adversarially trained EfficientNet-B0 with its regularly-trained counterpart (more comparisons are included in Appendix L). We observe that adversarial training leads to a characteristically different distribution of learned frequencies during training (Figure 4). Especially in the first layers, the network learns predominantly from low frequencies, which enables the network to preserve the global image content, rather than overfitting on high-frequency details such as texture. Interestingly, the adversarially-trained model learns this behavior in the early training stages, and faster than under normal training conditions (Rahaman et al., 2019). Deeper layers on the other hand show no salient differences.

Based on these findings, we propose a transformation approach of convolution weights into the frequency domain to interact with frequency information. Secondly, based on the latter finding we propose a high-frequency regularization, to further enforce the low-frequency bias in the first network layers and thus increase the native robustness.

## 4 Modifications to the convolution layers

Let us first formalize the computation flow in a conventional 2D convolution layer $f_{conv2d}(\boldsymbol{x}; \mathbf{W})$, $f_{conv2d}$ transforming an input signal $\boldsymbol{x}$ with $d_{in}$ input-channels into a signal with $d_{out}$ output-channels using a convolution kernel with a size of $k_0 \times k_1$. Further, let $\mathbf{W} \in \mathbb{R}^{d_{out} \times d_{in} \times k_0 \times k_1}$ denote the learned weights (*i.e.* the set of all kernels $\mathbf{W}_{i,j}$ in the respective layer, without bias). Without loss of generality, we assume $k_0 = k_1 = k$ in this paper. The output of $f_{conv2d}(\boldsymbol{x}; \mathbf{W})$ is then defined as:

$$\boldsymbol{y}_s = \sum_{d=0}^{d_{\text{in}}-1} \mathbf{W}_{s,d} * \boldsymbol{x}_d, \text{ for } s \in \{0, \ldots, d_{\text{out}}-1\}. \tag{3}$$

In the following, we propose a simple representation in the frequency space by replacing the convolution weight $\mathbf{W}$ with a combination of learned coefficients on the DCT-II basis. In this work, we limit ourselves to kernels with $k \geq 3$. We realize this by two common implementations seen in related literature (*e.g.* Ulicny et al. (2022)). Schematic visualizations of both approaches can be found in Appendix E.

**Weight decomposition (WD).** Our first approach decomposes the weight in a convolution layer into learnable coefficients $\mathbf{C} \in \mathbb{R}^{d_{out} \times d_{in} \times k \times k}$ and the basis $\mathbf{V}$ defined in Equation 2: $\mathbf{W} = \mathbf{C} \cdot \mathbf{V}$. Then, the

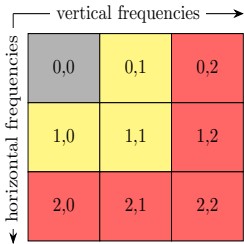

Figure 5: Regularization area of the coefficients in an individual $3 \times 3$ filter kernel. Colors match Equation 7.

$$\mathcal{R}(\mathbf{C}) = \underbrace{\|\mathbf{C}_{[s],[d],I,J}\|_2}_{\bullet} \cdot I +$$

$$\rho_{diff} \cdot \max(\underbrace{\|\mathbf{C}_{[s],[d],L,M}\|_2 - \overbrace{\|\mathbf{C}_{[s],[d],0,0}\|_2}^{\circ}}_{\circ}, 0), \qquad (7)$$

$$\text{for } I, J = \{\lceil k/2 \rceil, \ldots, k\}, L, M = \{0, 1\},$$
$$[s] = \{0, \ldots, d_{\text{out}} - 1\}, [d] = \{0, \ldots, d_{\text{in}} - 1\}.$$

convolution can be rewritten as:

$$\boldsymbol{y}_s = \sum_d (\mathbf{C}_{s,d} \cdot \mathbf{V}) * \boldsymbol{x}_d = \sum_{d,m,n} (\mathbf{C}_{s,d,m,n} \cdot \mathbf{V}_{m,n}) * \boldsymbol{x}_d. \qquad (4)$$

This increases the parameters to be kept in memory by a factor of 2 and adds one additional tensor multiplication per layer. However, these additional parameters are constant and do not need to be learned.

**Signal decomposition (SD).** Alternatively, our second approach does not replace the convolution weight $\mathbf{W}$ directly but performs a depthwise convolution of all combinations of inputs and the fixed basis vectors which is then aggregated by a learnable pointwise $(1 \times 1)$ convolution.

$$\boldsymbol{y}_s = \sum_{d,m,n} \mathbf{C}_{s,d,m,n} \cdot (\mathbf{V}_{m,n} * \boldsymbol{x}_d). \qquad (5)$$

This increases the parameter number by a factor of $d_{in}k^2$ to be kept in memory. Again, the number of learnable parameters is not increased. Also, note that the associativity property of convolution reveals the equivalence of both formulations in the forward pass:

$$\boldsymbol{y}_s = \sum_{d,m,n} \mathbf{C}_{s,d,m,n} \cdot (\mathbf{V}_{m,n} * \boldsymbol{x}_d) = \sum_{d,m,n} (\mathbf{C}_{s,d,m,n} \cdot \mathbf{V}_{m,n}) * \boldsymbol{x}_d. \qquad (6)$$

However, due to different learning dynamics, the modifications may converge to different solutions. In both approaches, the initial coefficient weights are sampled from a uniform distribution with an adjusted scale as per He et al. (2015). For the *weight decomposition* approach, we use $d_{in}k^2$ as fan information. The basis vectors are initialized as defined in Section 3 without any further adjustments.

### 4.1 Frequency coefficient regularization

As we have seen in Section 3.1 neural networks are biased towards low-frequency information, while early layers also introduce more magnitude on high frequencies. However, adversarial training increases the low-frequency bias already in the early training stages resulting in an overall low-frequency dominance after convergence in the first layers. To make use of this finding and increase the robustness of CNNs directly without adversarial training, we propose to regularize the DCT-II coefficients and explore the frequency shift and performance.

The proposed regularization (Equation 7 and Figure 5) regularizes the highest frequencies and additionally forces the first coefficient to have a higher magnitude than the subsequent frequency. Such behavior, that the weight coefficients decay with their corresponding frequencies, can also be observed in the adversarially robust weights in Figure 5. Interestingly, this is also in line with the theoretically expected behavior of frequency spectra that represent shapes rather than textures. We provide more theoretic background on this aspect in Appendix P. The occurrence of this latter constraint is determined by the binary hyperparameter $\rho_{diff}$, with $\rho_{diff} = 1$ throughout the paper, if not stated otherwise. The multiplicative term $\frac{k}{2}$ increases

penalization of higher frequencies. Let $coefs(\theta, h)$ denote a function that returns the set of convolution coefficient weights of the learnable parameters $\theta$ in the first $1/h$ section of the network depth. To enforce the dominance of low frequencies in early layers, we set $h = 3$ as our default value. We train the network with the following modifications to the objective:

$$\min_{\theta} \mathcal{L}\left(f\left(\boldsymbol{x}; \theta\right), \boldsymbol{y}\right) + \lambda \sum_{\mathbf{C} \in coefs(\theta, h)} \mathcal{R}(\mathbf{C}). \tag{8}$$

Where $\boldsymbol{x}, \boldsymbol{y} \sim \mathcal{D}$ denotes the training dataset and $\mathcal{L}$ is the original objective. An exemplary visualization of the learned coefficients under regularization is given in Figure 1 for $h \in \{1, 3\}$.

Table 1: **Performance evaluation** of various networks (ResNet-20/9 and EfficientNet-B0) on multiple datasets (CIFAR-10/100, SVHN, TinyImageNet, MNIST) before and after our applied regularization. We report clean accuracy on the respective test sets as well as adversarial robustness against FGSM, PGD-40, and AutoAttack for $L^{\infty}, \epsilon = 1/255$ ($\epsilon = 16/255$ for MNIST). We report averages over 5 runs.

**CIFAR-10**

| | Variant | Clean (↑) Val Acc. | Adversarial Acc. (↑) FGSM | PGD-40 | AA |
|---|---|---|---|---|---|
| ResNet-20 | CNN | 91.29 | 50.49 | 30.92 | 10.78 |
| | WD | 91.04 | 48.40 | 30.37 | 10.72 |
| | SD | **91.36** | 50.83 | 32.98 | 11.97 |
| | WD + Reg. | 89.86 | 50.85 | 41.81 | 26.79 |
| | SD + Reg. | 90.54 | **53.12** | **44.42** | **29.14** |
| ResNet-9 | CNN | **94.29** | 59.58 | 53.04 | 37.49 |
| | WD | 93.73 | 55.51 | 49.84 | 35.23 |
| | SD | 93.97 | 55.73 | 50.29 | 36.0 |
| | WD + Reg. | 93.18 | 59.25 | 56.08 | 43.62 |
| | SD + Reg. | 93.09 | **59.87** | **56.89** | **44.80** |
| Eff.Net-B0 | CNN | 90.38 | 53.55 | 54.05 | 45.51 |
| | WD | **90.51** | 49.87 | 49.97 | 40.76 |
| | SD | 90.44 | 51.04 | 51.77 | 43.39 |
| | WD + Reg. | 88.97 | **57.91** | 59.60 | 53.30 |
| | SD + Reg. | 89.18 | 57.83 | **59.68** | **53.50** |

| | Variant | Clean (↑) Val Acc. | Adversarial Acc. (↑) FGSM | PGD-40 | AA |
|---|---|---|---|---|---|
| CIFAR-100 ResNet-20 | CNN | **60.41** | 14.36 | 5.45 | 1.17 |
| | WD | 58.90 | 12.84 | 4.68 | 1.01 |
| | SD | 60.34 | 13.87 | 5.18 | 1.13 |
| | WD + Reg. | 56.65 | 16.85 | **12.73** | **5.59** |
| | SD + Reg. | 58.19 | **17.20** | 12.24 | 5.11 |
| SVHN ResNet-20 | CNN | 96.31 | 83.84 | 79.94 | 69.81 |
| | WD | **96.35** | 83.52 | 80.01 | 71.25 |
| | SD | 96.34 | 84.07 | 80.64 | 71.74 |
| | WD + Reg. | 96.28 | 84.11 | 81.21 | **73.27** |
| | SD + Reg. | 96.34 | **84.17** | **81.23** | 73.03 |
| TinyImNet ResNet-9 | CNN | **53.20** | 17.79 | 16.76 | 9.57 |
| | WD | 52.08 | 17.11 | 16.19 | 9.40 |
| | SD | 52.12 | 16.85 | 15.88 | 9.15 |
| | WD + Reg. | 51.25 | 18.10 | 17.34 | 10.23 |
| | SD + Reg. | 51.22 | **18.26** | **17.40** | **10.39** |
| MNIST ResNet-20 | CNN | 99.68 | 89.74 | 45.37 | 8.92 |
| | WD | 99.69 | 90.45 | 47.06 | 10.22 |
| | SD | 99.65 | **91.23** | 54.08 | 16.80 |
| | WD + Reg. | 99.69 | 90.70 | **55.84** | **25.92** |
| | SD + Reg. | 99.69 | 88.98 | 50.02 | 21.71 |

## 5 Experiments

In the following, we compare different architectures, with regular convolutions, and both decomposition variants (WD/SD) at varying frequency regularization (+ Reg.) (Equation 7). For each combination, we report results on clean accuracy, as well as robustness to various aspects.

**Models and datasets.** We evaluate low-resolution datasets such as CIFAR-10/100 (Krizhevsky, 2009), MNIST (LeCun et al., 2010), SVHN (Netzer et al., 2011), and Tiny-ImageNet (Le & Yang, 2015) on ResNet-20 (as introduced for CIFAR in He et al. (2016)), ResNet-9 - a regular and larger ResNet with optimization for CIFAR and a reduced number of layers (see Appendix D for architecture details), and an EfficientNet-B0 (Tan & Le, 2019) where we remove striding from the stem convolution. For ImageNet (Deng et al., 2009), we evaluate EfficientNet-B0 (Tan & Le, 2019) and ConvNeXt-Tiny (Liu et al., 2022). We test $h \in \{1, 3\}$ and $\lambda \in \{0.01, 0.05, 0.1\}$ and report results for the best performance over the mean of 5 runs except for ImageNet (1 run). Details regarding the training can be found in Appendix B.

Note that we have selected models with different kernel sizes - *e.g.* after the stem, ResNets use $k = 3$, EfficientNets-B0 mix $k = 3$ and $k = 5$, and ConvNeXts $k = 7$ (and $k = 2$ downsampling layers). The variance in kernel size allows us to demonstrate the transferability of our proposed regularization beyond the common $k = 3$ kernels.

Table 2: **Comparison against other robustness techniques** (Grabinski et al., 2022a; Lopes et al., 2019) of ResNet-20, ResNet-9, and EfficientNet-B0 on CIFAR-10. We report the mean clean validation accuracy and robust accuracy against adversarial attacks: FGSM, PGD-40, and AutoAttack for $L^\infty, \epsilon = 1/255$, and the mean corruption accuracy on CIFAR-10-C. We report averages over 5 runs.

| | Variant | Clean (↑) Val Acc. | Adversarial Acc. (↑) | | | CC (↑) Acc. |
|---|---|---|---|---|---|---|
| | | | FGSM | PGD-40 | AA | |
| ResNet-20 | CNN | 91.29 | 50.49 | 30.92 | 10.78 | 67.96 |
| | FLC (Grabinski et al., 2022a) | **91.52** | 52.49 | 30.25 | 8.48 | 68.75 |
| | PaGA (Lopes et al., 2019) | 91.29 | 50.36 | 31.50 | 11.38 | 67.73 |
| | Blur Pooling (Zhang, 2019) | 89.89 | 41.61 | 29.43 | 15.58 | 66.73 |
| | Adaptive Blur Pooling (Zou et al., 2020) | 89.48 | 41.94 | 32.09 | 18.22 | 67.17 |
| | Wavelet Pooling (Li et al., 2020) | 89.89 | 41.17 | 27.88 | 13.78 | 67.40 |
| | WD | 91.04 | 48.40 | 30.37 | 10.72 | 66.92 |
| | SD | 91.36 | 50.83 | 32.98 | 11.97 | 67.48 |
| | WD + Reg. | 89.86 | 50.85 | 41.81 | 26.79 | 74.04 |
| | SD + Reg. | 90.54 | **53.12** | **44.42** | 29.14 | **74.14** |
| ResNet-9 | CNN | 94.29 | 59.58 | 53.04 | 37.49 | 73.38 |
| | FLC (Grabinski et al., 2022a) | 94.24 | 59.64 | 53.47 | 38.65 | 73.81 |
| | PaGA (Lopes et al., 2019) | **94.33** | 59.12 | 52.62 | 37.50 | 73.72 |
| | WD | 93.73 | 55.51 | 49.84 | 35.23 | 72.87 |
| | SD | 93.97 | 55.73 | 50.29 | 36.00 | 73.48 |
| | WD + Reg. | 93.18 | 59.25 | 56.08 | 43.62 | 76.41 |
| | SD + Reg. | 93.09 | **59.87** | **56.89** | 44.80 | **77.72** |
| EfficientNet-B0 | CNN | 90.38 | 53.55 | 54.05 | 45.51 | 68.09 |
| | FLC (Grabinski et al., 2022a) | 89.68 | 51.92 | 53.09 | 45.37 | 69.72 |
| | PaGA (Lopes et al., 2019) | **90.72** | 54.18 | 54.97 | 46.64 | 69.31 |
| | WD | 90.51 | 49.87 | 49.97 | 40.76 | 67.10 |
| | SD | 90.44 | 51.04 | 51.77 | 43.39 | 66.65 |
| | WD + Reg. | 88.97 | **57.91** | 59.60 | 53.30 | **72.14** |
| | SD + Reg. | 89.18 | 57.83 | **59.68** | 53.50 | 71.87 |

Table 3: A **benchmark** of a ResNet-20 (CIFAR-10) with various architecturial frequency attenuation techniques evaluated with a batch size of 512 on an NVIDIA A100 GPU.

| Variant | Total Params (↓) | Learnable Params (↓) | Throughput (k img/sec) (↑) | Batch Update (ms) (↓) |
|---|---|---|---|---|
| CNN | 272.5k | 272.5k | 128.0 | 24.7 |
| FLC (Grabinski et al., 2022a) | 272.5k | 272.5k | 82.7 | 30.2 |
| Blur Pooling (Zhang, 2019) | 272.5k | 272.5k | 113.0 | 26.5 |
| Adaptive Blur Pooling (Zou et al., 2020) | 272.7k | 272.7k | 34.4 | 44.0 |
| Wavelet Pooling (Li et al., 2020) | 272.5k | 272.5k | 103.0 | 26.5 |
| WD (+ Reg.) | 274.0k | 272.5k | 124.3 | 24.9 |
| SD (+ Reg.) | 323.3k | 272.5k | 28.0 | 51.9 |

**Robustness evaluation.** To understand the effect on robustness and generalization of our proposed decomposition and regularization approaches, we run the standard AutoAttack test suite (AA) (Croce & Hein, 2020a) and additional FGSM-, and PGD-attacks at $\epsilon = 1/255$ ($\epsilon = 16/255$ for MNIST) under the $L^\infty$-norm. We use *Foolbox* (Rauber et al., 2017) to run both FGSM and PGD at the default setting (*e.g.* 40 steps for PGD). We do not include AA results for ImageNet, as these models barely withstand any attacks and measure robust accuracies of 0% even at this small $\epsilon$ without adversarial training. Further, we evaluate the robustness of common corruptions of CIFAR-10 and ImageNet models on the respective corrupted datasets (Hendrycks & Dietterich, 2019). In addition, we are interested in the behavior of the methods towards texture bias (Geirhos et al., 2019) and OOD generalization tests (Geirhos et al., 2021). Hence, we evaluate our ImageNet (Deng et al., 2009) models on 5 of these OOD datasets: texture-shape cue-conflict, ImageNet-Sketch, Stylized-ImageNet, and edge-/silhouette-transformations of ImageNet using the implementation of Geirhos et al. (2021).

## 5.1 Low-resolution datasets

**CIFAR-10.** As to be expected, switching from regular to either decomposition variant has an insignificant impact on the clean accuracy and a small effect on robust accuracy (at all adversarial attacks) (Table 1).

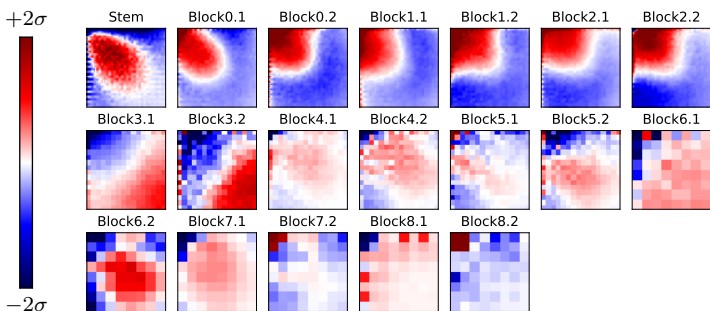

Figure 6: Frequency heat maps of **feature maps**: Regularized layers (top row) of a ResNet-20 show a shift towards LF after regularization, compared to non-regularized layers in rows two and three.

However, applying the regularization clearly improves robustness towards all attacks, while slightly decreasing clean accuracy. We can also observe that SD slightly outperforms WD on almost all tested architectures. Hence, it may be tempting to only proceed with SD. However, the additionally created channels account for more parameters, a large memory overhead, and slower inference and training performance. *E.g.* on ResNet-20 we see a 4.4x slower forward pass and 18% more total parameters, while WD has a minimal overhead, both, in parameters and throughput (Table 3).

Regarding robustness, we see the largest gains on models that initially performed worst (+18.36% on ResNet-20 on CIFAR-10). Out of all our tested models, EfficientNet-B0 is the most robust, both, before and after regularization. Noticeably, even the worst hyperparameter combination for ResNet-20 (WD, $\lambda = 0.01, h = 1, \rho_{diff} = 0$) still achieves a 14.22% higher AA accuracy than the baseline. A complete overview of tested hyperparameters is given in Appendix F.

**Common corruptions of CIFAR-10.** For common corruptions (CC) (the last column in Table 2 corresponds to the CIFAR-10 results in Table 1), we analyze the mean accuracy over all corruptions and severities, as well as individual results for corruptions at the highest severity level. A complete overview is given in Appendix C. Similar to the results on adversarial robustness we observe that on average both regularized variants outperform the baseline. Additionally, regularized models become significantly more robust against corruptions having predominantly high frequency (HF) perturbations (see Yin et al. (2019) for spectrums) such as *pixelate* and *defocus/glass/gaussian blur*. Perhaps less surprisingly, regularized models become less sensitive to increased *JPEG compression*, as they rely on (quantized) DCT-II coefficients. For corruptions with larger variance in the frequency spectrum, regularized performance remains largely unchanged. We see a slight degradation of performance in low frequency (LF) corruptions such as *brightness*, *saturation*, *contrast*, and *impulse noise*. However, the accuracy drop is relatively low considering the evaluation at the highest severity level.

**Other datasets.** Although several works reported a shift in the frequency band of adversarial attacks depending on the dataset (Maiya et al., 2021; Abello et al., 2021; Bernhard et al., 2021; Ortiz-Jiménez et al., 2020), we consequently see an improvement due to our HF regularization on multiple datasets (Table 1). Arguably, we see smaller improvements for SVHN/Tiny-ImageNet - which are also the datasets that show more LF perturbations than HF. Contrary to our CIFAR-10 results, WD outperforms SD on all datasets except SVHN.

**Spectrum of feature maps.** Further, we aim to understand the implications of the regularization on the computed feature maps. Exemplarily, we compare an *SD + Reg.* ResNet-20 against a CNN baseline and analyze the magnitude shift in the DCT-II coefficients of the feature maps (Figure 6) of a clean validation batch. Our regularization causes a clear shift towards lower frequencies in regularized layers. Interestingly, in the stem layer, we also see large shifts from entirely vertical or horizontal frequencies to more balanced ones. Contrary, non-regularized (deeper) layers appear to slightly shift towards higher frequencies.

Table 4: Results on **ImageNet** for EfficientNet-B0 and ConvNeXt-Tiny on clean data, FGSM, PGD-40 ($L^\infty, \epsilon = 1/255$), ImageNet-C and out-of-distribution generalization test. All regularization hyperparameters are $\lambda = 0.05$ and $h = 3$.

| | Variant | Clean Val. Acc. (↑) | | Adversarial Attacks Acc. (↑) | | Corruption Error (↓) | Cue-Conflict (↑) | Sketch (↑) | | Stylized (↑) | | Edge (↑) | Silhouette (↑) |
|---|---|---|---|---|---|---|---|---|---|---|---|---|---|
| | | Top 1 | Top 5 | FGSM | PGD-40 | ImageNet-C | | Top 1 | Top 5 | Top 1 | Top 5 | Top 1 | Top 1 |
| EfficientNet-B0 | CNN | 75.44 | 92.86 | 16.89 | 2.32 | 54.54 | 23.52 | 65.25 | 84.62 | 52.25 | 79.00 | 35.00 | 51.25 |
| | WD | 75.80 | 92.94 | 14.86 | 2.03 | 53.99 | 22.58 | 66.12 | 86.25 | 48.25 | 78.88 | 40.62 | 55.00 |
| | SD | 75.62 | 92.82 | 15.05 | 1.41 | 52.85 | 23.67 | 66.38 | 84.50 | 52.50 | 78.50 | 34.38 | 58.13 |
| | WD + Reg. | 75.44 | 92.15 | 18.45 | 4.43 | 52.03 | 29.38 | 66.38 | 86.88 | 47.62 | 79.75 | 36.25 | 58.13 |
| | SD + Reg. | 74.42 | 92.19 | 18.70 | 5.33 | 51.12 | 25.78 | 64.75 | 87.88 | 49.12 | 77.12 | 32.50 | 58.75 |
| ConvNeXt-Tiny | CNN | 81.32 | 95.53 | 35.53 | 3.93 | 41.92 | 24.84 | 71.50 | 88.00 | 56.00 | 78.38 | 48.12 | 62.50 |
| | WD | 81.11 | 95.55 | 35.30 | 2.97 | 42.98 | 25.31 | 73.12 | 89.62 | 52.00 | 77.62 | 47.50 | 58.75 |
| | WD + Reg. | 79.25 | 94.38 | 35.69 | 4.22 | 44.31 | 32.27 | 73.75 | 88.12 | 58.00 | 82.38 | 38.75 | 65.62 |

Table 5: **FGSM-Adversarial Training on CIFAR-10** with $L^\infty$, $\epsilon = 8/255$. We report the mean over 5 runs for the FGSM train and validation accuracy of the epoch of the best PGD-40 validation accuracy as well as the AutoAttack accuracy. We also report the corresponding mean accuracy on CIFAR-10-C and report the difference to the clean-trained evaluations.

| | Variant | Clean Val. Acc. (↑) | FGSM (↑) Train Acc. | Adversarial Acc. (↑) | | Corruption Acc. | |
|---|---|---|---|---|---|---|---|
| | | | | PGD-40 | AA | Mean (↑) | Δ (AT-Normal) (↑) |
| ResNet-20 | CNN | **73.73** | **50.39** | 46.14 | 36.09 | **66.99** | -0.97 |
| | WD + Reg. | 71.69 | 48.46 | 45.38 | 35.64 | 65.48 | -8.56 |
| | SD + Reg. | 73.01 | 49.93 | **46.34** | **36.47** | 66.73 | -7.41 |
| ResNet-9 | CNN | 81.70 | 60.27 | **52.77** | 0.00 | 74.06 | 0.68 |
| | WD + Reg. | 81.56 | 61.66 | 51.52 | 39.97 | 74.47 | -1.94 |
| | SD + Reg. | **82.56** | **63.39** | 51.80 | **40.14** | **75.40** | -2.32 |
| EfficientNet-B0 | CNN | 63.00 | 42.34 | 42.49 | 34.04 | 57.35 | -10.74 |
| | WD + Reg. | 68.50 | 45.87 | 45.13 | 36.56 | 62.66 | -9.48 |
| | SD + Reg. | **68.89** | **46.76** | **45.57** | **36.76** | **63.07** | -8.8 |

Table 6: **PGD-Adversarial Training on ImageNet** of ResNet-50 with $L^\infty$, $\epsilon = 4/255$. We report the train and validation accuracy under PGD attacks, validation accuracy under AutoAttack, corruption error, and cue-conflict. Results are from one run.

| Variant | Clean Val Acc. (↑) | PGD (↑) Train Acc. | Adversarial Acc. (↑) | | Corruption Error (↓) | Cue-Conflict (↑) |
|---|---|---|---|---|---|---|
| | | | PGD | AA | | |
| CNN | 56.85 | 33.88 | 36.04 | 22.33 | 78.65 | 38.83 |
| WD | 55.82 | 33.91 | 35.06 | 22.06 | 78.90 | 38.83 |
| WD + Reg. | **58.09** | **36.00** | **37.09** | **24.32** | **78.36** | **39.38** |

**Comparison to other methods.** We compare our method to *FrequencyLowCut Pooling (FLC)* (Grabinski et al., 2022a), *Patch Gaussian Augmentation (PaGA)* (Lopes et al., 2019), and, on ResNet-20, *Blur Pooling* (Zhang, 2019), *Adaptive Blur Pooling* (Zou et al., 2020), and *Wavelet Pooling* (Li et al., 2020) (Table 2) as these methods also aim at HF-regularization. Regarding AA and CC performance, we observe that our method consistently outperforms these other approaches in standalone comparisons with small degradation of clean validation accuracy. Additionally, imposing our regularization on top of other methods can improve their robustness significantly, as we show for FLC and PaGA in Appendix I. Interestingly, we often get the highest levels of robustness in combination with another method, proving that our regularization can be complementary to other robustness techniques. For a comparison to Wang et al. (2020), please refer to Appendix I, where we show favorable behavior of our approach.

## 5.2 ImageNet

Next, we aim to explore how our regularization performs on the common ImageNet dataset (Deng et al., 2009). In particular, more OOD tests exist for this dataset which allows us to study aspects outside adversarial robustness, and robustness against common corruptions. Similar to our results on other datasets, we see an improvement in adversarial robustness at slight (1-2%) degradation of clean performance (Table 4). While we see an improvement in CC performance on EfficientNet, we see an equal decrease for ConvNeXt. This may be due to the larger kernels ($7 \times 7$) that ConvNeXt utilizes and may, thus, require other hyper-

parameters. Importantly we see a significant improvement of the *cue-conflict* in both cases - which is also reflected in the increased accuracy of *silhouette* (LF) and the decrease in performance of *edge* (HF). This indicates that our regularization favorably shifts models toward shape bias (Geirhos et al., 2019).

### 5.3 Integration into adversarial training and impact on robust overfitting

So far, we investigated the effect of our proposed HF regularization on native robustness. For completeness, we aim to explore the role of our regularization in AT. We train our models against FGSM-adversaries on CIFAR-10 ($L^\infty, \epsilon = 8/255$) (Table 5). To avoid robust overfitting, we use early stopping based on PGD-40 test performance. We observe, that our regularization has a beneficial effect on the out-of-domain attacks (*i.e.* AA) and all runs show an increased performance after regularization. We furthermore observe that our regularization appears to mitigate *robust overfitting* of training attacks (similarly to Grabinski et al. (2022a)) on ResNet-9: without regularization the AT-trained CNN achieves high FGSM train accuracy and high PGD-40 validation accuracy but fails to generalize to other attacks (AA) and stagnates at 0%. With regularization, all runs show comparable or even better accuracy than the best non-regularized models. However, similarly to the observations by Saikia et al. (2021), we generally see a significant decrease in CC accuracy due to AT. Again, this demonstrates that AT is not the cure-all to improve network robustness and there is a need for other approaches such as our proposed frequency regularization.

Additionally, we extend our experiments to AT on ImageNet (Table 6). Here we switch to single-step PGD training with the common $\epsilon = 4/255$ and train the ResNet-50 architecture. Again we report PGD, AA, and CC performance but this time also the cue-conflict score. Our regularized WD architecture outperforms the baseline in all metrics: adversarial robustness, corruption error, and cue-conflict. This demonstrates that our method can mitigate some of the overfitting aspects of adversarial training and leads to an improved OOD generalization performance. We do not report the (regularized) SD performance due to the lack of tuned hyperparameters but expect similar gains when tuned properly.

## 6 Conclusion

We have shown a first step towards improving the native robustness of CNNs to multiple distribution shifts such as adversarial attacks, corruptions, and shape-biased datasets, as well as the benefit of our regularization for adversarial training. In particular, our regularization decreases the sensitivity to high-frequency perturbations. Albeit our results do not approach SOTA levels, we emphasize that we improve robustness on a wide range of tests, whereas SOTA methods like AT often overfit to one specific type of robustness, such as adversarial attacks, and often even impair performance on other tests compared to normal baselines. Additionally, our method does not rely on OOD examples but intrinsically strengthens the model. Our approach has shown to generalize to different networks with various kernel sizes, that were trained on different datasets, and different measures of robustness. We have also shown that our method can be used in combination with other approaches such as *PaGA* (Lopes et al., 2019), *FLC* (Grabinski et al., 2022a), and even AT (Madry et al., 2018) to further improve robust performance. In combination with AT, our approach shows promise to mitigate *robust overfitting* (Rice et al., 2020).

**Limitations.** We observed that on some architectures switching to WD/SD introduces a significant drop in accuracy (before regularization). Although the forward pass of both methods is mathematically equivalent to baselines, the backward pass is not. *E.g.* weight updates on linear combinations of decomposed convolution filters and feature maps are in different backward pass stages and under different quantization conditions due to limited bit precision. While we observe that our regularization generally improves a multitude of robustness aspects, the regularized counterparts may underperform CNN baselines due to the initial impairment due to the architecture change. We aim to explore more root causes and alternatives in future work.

### Acknowledgments

JL and MK acknowledge support from the DFG research unit 5336 "Learning to Sense".

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

**Organization of the Appendix.** In the following, we provide additional information and details that accompany the main paper. We first describe the broader impact of our method. Secondly, describe the details of training setups for the different models and datasets. We give more information on the results of common corruptions, *i.e.* CIFAR-10-C. We provide an overview of the used ResNet-9 architecture. Further, we visualize both our proposed decomposition approaches, *weight decomposition* and *signal decomposition*. Then, we include all hyperparameter exploration results for both our methods and further motivate our regularization approach by ablating two other regularization types. We show results of adversarial training on ImageNet and additional comparisons to other robustness techniques and show the impact of corruptions and adversarial samples in the frequency domain. Also, we include details on the shape bias visualization. Next, we provide detailed plots on the coefficient distributions of clean-trained and adversarial-trained networks, *i.e.* ResNet-20, ResNet-9, and EfficientNet-B0, over all layers and epochs. Then, we include a detailed version of Table 1 in the main paper. We visualize and interpret the attributions before and after regularization, and show the Fourier spectrum of activations (and their differences). Finally, we provide a theoretical motivation for our proposed regularization based on the frequency spectra of shape signals.

- Broader impact in Appendix A

- Training details in Appendix B

- Details on corruption experiments in Appendix C

- ResNet-9 architecture in Appendix D

- Architecture diagram of *weight/signal decomposition* in Appendix E

- Hyperparameter exploration in Appendix F

- Adversarial training on ImageNet in Appendix G

- Ablation with other regularization approaches in Appendix H

- Comparison to other robustness techniques in Appendix I

- Analysis of corruptions and adversarial samples in the frequency domain in Appendix J

- Details on shape bias in Appendix K

- Evolution plots in Appendix L

- Detailed table including standard deviation in Appendix M

- Qualitative analysis of attribution maps in Appendix N

- Fourier spectrum of activations in Appendix O

- Background on the link between shape and low-frequency bias as motivation for our proposed regularization in Appendix P

## A  Broader Impact

- **Arms race:** Adversarial training, which is a common technique to enhance model robustness, involves introducing worst-case perturbations into the training data. However, the very same techniques used for adversarial training can also be employed by malicious actors to craft adversarial attacks. As AI models become more robust to conventional adversarial attacks, adversaries may develop more sophisticated and potent attack strategies. This arms race in adversarial techniques could potentially lead to an escalation of cyber threats, with negative consequences for cybersecurity and data privacy.

- **Native robustness:** In theory, one could attempt to characterize all possible perturbations of train data and perform adversarial training on those. However, characterizing all possible perturbations is already very hard for specific problems and potentially insolvable for vision in general (e.g., all the possible perturbations that could happen in autonomous driving). This problem hasn't been solved to date. Further, even if we were able to characterize all possible perturbations, we could only train on a discrete set, which could only approximate the continuous search space and, thus, risk overfitting. Lastly, to effectively integrate this into current training approaches, we would have to significantly increase the training time to compensate for the new data. In contrast, by native robustness, we seek to add a low-frequency prior to the network to steer the training to a more generalizable representation without additional (perturbated/augmented) data. Still, we remain compatible with these methods and even adversarial training. In Section 5.3 we even show a reduced risk of robust overfitting after regularization.

- **Ethical and bias:** While the paper's approach to prioritizing low-frequency information in decision-making can mitigate certain types of bias, it is crucial to recognize that bias is generally a multi-faceted issue. Shifting the focus toward low-frequency information may introduce its own set of biases. For example, in computer vision, prioritizing shapes might lead to underrepresentation or misrepresentation of fine-grained details, which can have implications in fields like medical diagnostics and object recognition. It is essential to carefully consider and address these potential biases to ensure fair and equitable AI systems.

## B    Training details

### B.1    Low resolution: CIFAR-10/100, MNIST, SVHN, Tiny-Imagenet

**Training setup.**   We train models for all low-resolution datasets with the same hyperparameters. Models are trained for 120 epochs. For both ResNets we use an SGD optimizer (with Nesterov momentum of 0.9) with an initial learning rate of 1e-2 that we downscale by 0.1 every 30 epochs, and a weight decay of 1e-2. For EfficientNet-B0 we use an AdamW (Loshchilov & Hutter, 2019) optimizer with an initial learning rate of 1e-4 that follows a cosine annealing schedule, and a weight decay of 5e-2. In all cases, we use a batch size of 256, and categorical cross entropy as loss function with our regularization. We analyze models with weights learned after the last gradient update.

We use the following augmentations for datasets:

- **CIFAR-10/100:** Training images are zero-padded by 4 px along each dimension, apply random horizontal flips, and proceed with $32 \times 32$ px random crops. Test images are not modified.

- **Tiny-ImageNet:** Training images are obtained using randomly resized $56 \times 56$ px crops. Test images are $56 \times 56$ px center crops.

- **MNIST:** Train and test images are upscaled to $32 \times 32$ px.

- **SVHN:** Train and test images are not modified.

For all datasets, samples are normalized by the channel mean and standard deviation.

### B.2    ImageNet.

We train all ImageNet models with the default hyperparameters and augmentations for ConvNeXt-Tiny (Liu et al., 2022). In particular, we train 300 epochs with an effective batch size of 4096. For EfficientNet-B0, we reduce the batch size to 1024 due to memory constraints. Again, we evaluate model parameters learned after the last gradient update.

## C CIFAR10-C details

In this section we provide detailed information about the the corruption results on CIFAR-10-C (Hendrycks & Dietterich, 2019) in Table 7.

Table 7: Results (in % and averaged over 5 runs) on **CIFAR-10-C** for ResNet-20, ResNet-9 and EffcientNet-B0. Reported is the mean over all severities and corruptions and all corruptions at severity level 5.

| | Variant | Mean (↑) Accuracy | Noise Gaussian | Shot | Impulse | Blur Defocus | Glass | Motion | Zoom | Weather Snow | Frost | Fog | Brightness |
|---|---|---|---|---|---|---|---|---|---|---|---|---|---|
| ResNet-20 | CNN | 67.96 | 23.77 | 28.64 | **32.01** | 51.67 | 38.31 | 56.85 | 55.99 | 65.77 | 50.60 | 62.74 | 85.54 |
| | WD | 66.92 | 20.06 | 24.29 | 27.62 | 54.29 | 37.42 | 57.75 | 57.42 | 64.38 | 49.65 | 62.08 | 85.63 |
| | SD | 67.48 | 20.34 | 24.60 | 27.09 | 52.37 | 37.85 | 56.75 | 56.62 | 66.24 | 51.23 | **63.66** | **85.96** |
| | WD + Reg. ($\lambda = 0.05, h = 3$) | 74.04 | **34.40** | **39.78** | 25.73 | **64.69** | **55.87** | **64.51** | 70.84 | 73.93 | **70.38** | 60.49 | 84.52 |
| | SD + Reg. ($\lambda = 0.01, h = 3$) | **74.14** | 31.95 | 37.26 | 25.09 | 63.91 | 54.24 | 64.41 | **71.05** | **74.28** | 69.51 | 62.38 | 85.17 |
| ResNet-9 | CNN | 73.38 | 27.16 | 34.61 | 24.92 | 53.98 | 47.89 | 64.43 | 62.64 | 75.08 | 63.46 | 67.74 | **89.48** |
| | WD | 72.87 | 27.06 | 34.06 | 24.91 | 56.81 | 48.00 | 65.72 | 64.45 | 74.48 | 63.18 | 67.67 | 88.75 |
| | SD | 73.48 | 28.36 | 35.52 | **27.47** | 57.91 | 48.10 | 65.93 | 65.20 | 75.29 | 64.31 | 68.32 | 88.89 |
| | WD + Reg. ($\lambda = 0.01, h = 3$) | 76.41 | 32.71 | 38.94 | 23.21 | 64.86 | 60.63 | 67.13 | 71.71 | 79.42 | 75.45 | 66.25 | 87.85 |
| | SD + Reg. ($\lambda = 0.01, h = 3$) | **77.72** | **36.31** | **42.39** | 24.34 | **67.38** | **64.71** | **68.65** | **73.88** | **80.68** | **78.43** | 65.73 | 87.93 |
| EfficientNet-B0 | CNN | 68.09 | 25.02 | 29.04 | **29.41** | 45.76 | 47.17 | 52.34 | 53.14 | 65.74 | 54.80 | 54.81 | **84.06** |
| | WD | 67.10 | 18.56 | 22.33 | 23.02 | 48.38 | 44.17 | 55.67 | 55.24 | 67.11 | 56.59 | **58.52** | 83.96 |
| | SD | 66.65 | 17.59 | 21.15 | 21.70 | 47.46 | 44.97 | 55.08 | 55.47 | 65.41 | 53.16 | 57.39 | 84.02 |
| | WD + Reg. ($\lambda = 0.01, h = 3$) | **72.14** | **28.06** | **33.15** | 21.30 | **61.23** | 59.35 | **62.72** | **66.69** | **73.79** | 68.62 | 52.39 | 83.10 |
| | SD + Reg. ($\lambda = 0.01, h = 3$) | 71.87 | 26.99 | 32.48 | 20.52 | 59.29 | **61.23** | 59.11 | 65.19 | 73.60 | **69.59** | 50.22 | 83.01 |

| | Variant | Digital Contrast | Elastic Transform | Pixelate | JPEG Compression | Extra Speckle Noise | Gaussian Blur | Spatter | Saturate |
|---|---|---|---|---|---|---|---|---|---|
| ResNet-20 | CNN | 26.18 | 64.51 | 38.11 | 67.11 | 34.39 | 35.35 | 69.75 | **82.37** |
| | WD | 25.54 | 64.21 | 37.42 | 66.84 | 29.57 | 39.21 | 70.87 | 80.97 |
| | SD | **27.46** | 65.62 | 36.73 | 67.54 | 29.95 | 37.10 | 70.41 | 81.69 |
| | WD + Reg. ($\lambda = 0.05, h = 3$) | 23.69 | 71.65 | **81.36** | 76.15 | **43.27** | 58.97 | 74.83 | 78.53 |
| | SD + Reg. ($\lambda = 0.01, h = 3$) | 22.10 | 71.43 | 80.04 | **76.23** | 41.07 | **57.60** | **75.24** | 79.91 |
| ResNet-9 | CNN | 26.25 | 75.26 | 50.51 | 74.78 | 40.95 | 41.70 | 72.83 | **83.84** |
| | WD | 25.62 | 75.09 | 49.04 | 73.59 | 40.44 | 45.16 | 73.29 | 82.65 |
| | SD | **26.72** | 75.09 | 49.40 | 73.72 | 41.84 | 45.90 | 73.64 | 83.17 |
| | WD + Reg. ($\lambda = 0.01, h = 3$) | 24.10 | 78.52 | 77.45 | 79.45 | 42.89 | 56.68 | 79.25 | 81.15 |
| | SD + Reg. ($\lambda = 0.01, h = 3$) | 24.24 | **78.58** | **81.09** | **80.67** | **45.90** | **59.81** | **80.49** | 81.28 |
| EfficientNet-B0 | CNN | 20.46 | 70.68 | 47.06 | 73.33 | 33.99 | 35.53 | 73.88 | **81.92** |
| | WD | 19.34 | 69.90 | 47.27 | 72.11 | 28.27 | 37.72 | 72.20 | 80.48 |
| | SD | **22.23** | 71.09 | 46.38 | 72.29 | 26.91 | 37.71 | 73.91 | 81.25 |
| | WD + Reg. ($\lambda = 0.01, h = 3$) | 18.75 | 73.97 | 76.93 | **77.89** | **37.34** | 54.08 | 76.21 | 78.72 |
| | SD + Reg. ($\lambda = 0.01, h = 3$) | 17.37 | **74.41** | **78.30** | 77.85 | 36.56 | 51.89 | **76.72** | 78.10 |

# D    ResNet-9 architecture

In this section, we show the architectural topology of the ResNet-9 (Figure 7), used in the main paper in Section 5.

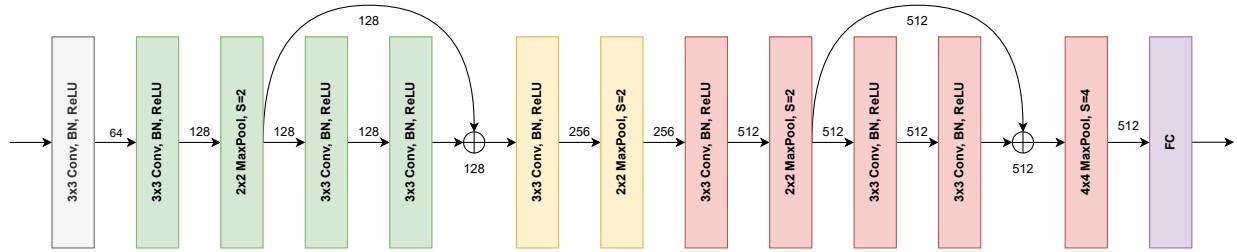

Figure 7: Architecture of ResNet-9. The numbers on the arrows indicate the number of channels. The initial and final number of channels depends on input data and the number of classes.

# E    Decompositions

In the following, we visualize both convolution decomposition approaches as presented in Section 4. Figure 8 shows the *weight decomposition* approach and Figure 9 the *signal decomposition* approach.

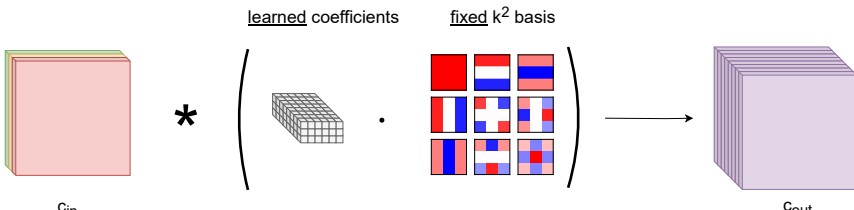

Figure 8: Flow of the *weight decomposition* implementation. Instead of learning the weight directly, we learn coefficients of a DCT-II basis and construct the weight via a linear combination.

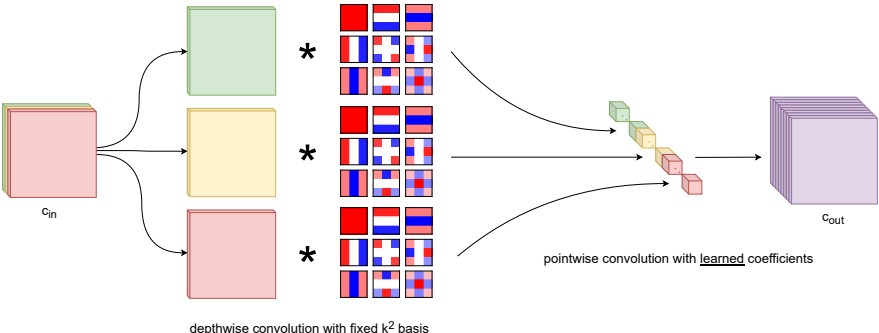

Figure 9: Flow of the *signal decomposition* implementation. Each input channel is convolved with all basis vectors in a depthwise convolution layer. The outputs are then aggregated by a pointwise $(1 \times 1)$ convolution.

## F  Hyperparameter exploration

In the main paper, we only included the results using $h = 3$ as regularization depth and $\lambda = 0.05$. For the sake of completeness, we further provide additional results on other regularization hyperparameters. In the following, we compare different regularization hyperparameters and evaluate them on the clean accuracy in CIFAR-10 (Krizhevsky, 2009) and against adversarial attacks, FGSM (Goodfellow et al., 2015), PGD-40 (Kurakin et al., 2017) and AutoAttack (Croce & Hein, 2020a) as well as against corruptions in CIFAR-10-C (Hendrycks & Dietterich, 2019). The evaluation setting is the same as in the main paper in Section 5. Table 8 presents an overview of different regularization depths for ResNet-20 (He et al., 2016), from regularizing only the first eighth layers (the first 3 layers in this case, $h = 8$) to regularizing the complete network (h=1). Note, $h = 2$ regularizes half of the network. As we can see, regularizing more than half of the network leads to a decrease in clean validation accuracy, with no overall benefit to robust accuracy. Whereas regularizing only within the first half of the network leads to the trade-off of slightly decreasing the clean accuracy compared to the non-regularized versions, but substantially increasing the robust accuracy, which we aim for in this paper. Also, our default setting in the main paper ($h = 3$) leads overall to the best robust accuracy.

In Table 9 and Table 10 we provide results on CIFAR-10 and CIFAR-100 for ResNet-20, ResNet-9 (He et al., 2016) and EfficientNet-B0 (Tan & Le, 2019) with different regularization hyperparameters: $\lambda$ as the loss parameterization Equation 8, has the network depth regularization (first $1/h$ is regularized) and the binary parameter $\rho_{\text{diff}}$ Equation 7. The evaluation on clean accuracy and adversarial robustness in the same as in the main paper in Section 5.

Table 8: Ablation of **ResNet-20** with **different regularization depths** on CIFAR-10 and evaluation against adversarial attacks, FGSM, PGD-40, AutoAttack, and corruption CIFAR-10-C. Reported is the mean accuracy in % and standard deviation over 5 runs.

| Variant | Clean Val Acc.($\uparrow$) | FGSM $\epsilon = 1/255$ ($\uparrow$) | PGD-40 $\epsilon = 1/255$ ($\uparrow$) | AA $\epsilon = 1/255$($\uparrow$) | Mean Accuracy CIFAR-10-C ($\uparrow$) |
|---|---|---|---|---|---|
| CNN | $91.29 \pm 0.14$ | $50.49 \pm 0.62$ | $30.92 \pm 0.88$ | $10.78 \pm 0.33$ | $67.96$ |
| WD | $91.04 \pm 0.21$ | $48.40 \pm 0.36$ | $30.37 \pm 0.98$ | $10.72 \pm 0.91$ | $66.92$ |
| SD | $\mathbf{91.36} \pm 0.23$ | $50.83 \pm 0.49$ | $32.98 \pm 1.06$ | $11.97 \pm 0.92$ | $67.48$ |
| WD + Reg. ($\lambda = 0.05, h = 3$) | $89.86 \pm 0.14$ | $50.85 \pm 0.71$ | $41.81 \pm 1.15$ | $26.79 \pm 1.52$ | $74.04$ |
| SD + Reg. ($\lambda = 0.05, h = 3$) | $90.34 \pm 0.14$ | $\mathbf{53.07} \pm \mathbf{0.29}$ | $\mathbf{44.04} \pm \mathbf{0.56}$ | $28.62 \pm 0.96$ | $73.98$ |
| WD + Reg. ($\lambda = 0.05, h = 1$) | $88.84 \pm 0.08$ | $48.99 \pm 0.90$ | $40.67 \pm 1.34$ | $26.19 \pm 1.52$ | $73.96$ |
| WD + Reg. ($\lambda = 0.05, h = 1.2$) | $88.78 \pm 0.21$ | $48.46 \pm 0.57$ | $39.99 \pm 0.86$ | $25.77 \pm 1.30$ | $73.56$ |
| WD + Reg. ($\lambda = 0.05, h = 1.4$) | $88.78 \pm 0.47$ | $48.42 \pm 1.30$ | $40.05 \pm 1.82$ | $25.81 \pm 2.20$ | $73.51$ |
| WD + Reg. ($\lambda = 0.05, h = 1.6$) | $88.77 \pm 0.32$ | $48.58 \pm 0.94$ | $40.57 \pm 1.22$ | $26.20 \pm 1.09$ | $73.94$ |
| WD + Reg. ($\lambda = 0.05, h = 1.8$) | $88.93 \pm 0.32$ | $48.42 \pm 0.90$ | $39.42 \pm 1.58$ | $24.74 \pm 2.55$ | $73.39$ |
| WD + Reg. ($\lambda = 0.05, h = 2$) | $88.99 \pm 0.22$ | $49.20 \pm 0.34$ | $40.54 \pm 0.70$ | $26.21 \pm 1.31$ | $74.14$ |
| WD + Reg. ($\lambda = 0.05, h = 4$) | $90.13 \pm 0.10$ | $51.20 \pm 0.75$ | $41.85 \pm 1.15$ | $26.08 \pm 1.30$ | $72.05$ |
| WD + Reg. ($\lambda = 0.05, h = 6$) | $90.05 \pm 0.31$ | $50.87 \pm 0.52$ | $41.17 \pm 0.86$ | $25.06 \pm 0.69$ | $71.79$ |
| WD + Reg. ($\lambda = 0.05, h = 8$) | $90.42 \pm 0.21$ | $50.45 \pm 0.35$ | $39.68 \pm 0.89$ | $22.82 \pm 1.43$ | $70.88$ |
| SD + Reg. ($\lambda = 0.05, h = 1$) | $89.26 \pm 0.32$ | $49.48 \pm 0.35$ | $41.23 \pm 0.43$ | $26.61 \pm 0.60$ | $74.32$ |
| SD + Reg. ($\lambda = 0.05, h = 1.2$) | $89.09 \pm 0.21$ | $50.11 \pm 0.74$ | $42.10 \pm 1.14$ | $27.75 \pm 1.86$ | $74.35$ |
| SD + Reg. ($\lambda = 0.05, h = 1.4$) | $89.28 \pm 0.22$ | $50.54 \pm 0.53$ | $42.19 \pm 1.06$ | $27.71 \pm 1.29$ | $74.34$ |
| SD + Reg. ($\lambda = 0.05, h = 1.6$) | $89.28 \pm 0.27$ | $51.03 \pm 1.56$ | $43.18 \pm 2.30$ | $\mathbf{28.92} \pm \mathbf{2.92}$ | $74.44$ |
| SD + Reg. ($\lambda = 0.05, h = 1.8$) | $89.52 \pm 0.34$ | $51.25 \pm 0.95$ | $43.04 \pm 1.49$ | $28.34 \pm 2.09$ | $74.67$ |
| SD + Reg. ($\lambda = 0.05, h = 2$) | $89.32 \pm 0.13$ | $50.90 \pm 0.68$ | $42.60 \pm 1.34$ | $27.75 \pm 1.68$ | $\mathbf{74.74}$ |
| SD + Reg. ($\lambda = 0.05, h = 4$) | $90.58 \pm 0.17$ | $52.41 \pm 0.69$ | $42.16 \pm 0.86$ | $25.53 \pm 1.17$ | $71.77$ |
| SD + Reg. ($\lambda = 0.05, h = 6$) | $90.61 \pm 0.25$ | $52.35 \pm 0.63$ | $42.46 \pm 0.77$ | $25.89 \pm 0.63$ | $71.88$ |
| SD + Reg. ($\lambda = 0.05, h = 8$) | $90.63 \pm 0.05$ | $51.92 \pm 0.98$ | $40.82 \pm 1.21$ | $23.43 \pm 1.18$ | $71.15$ |

Table 9: Ablation Results (in % and averaged over 5 runs) of ResNet-20, ResNet-9, and EfficientNet-B0 trained with different convolution implementations and varying regularization on **CIFAR-10**, and evaluation against adversarial attacks, FGSM, PGD-40 and AutoAttack and corruptions in CIFAR-10-C.

| | Variant | Clean Val Acc. in (%)($\uparrow$) | FGSM $\epsilon = 1/255$ ($\uparrow$) | PGD-40 $\epsilon = 1/255$ ($\uparrow$) | AA $\epsilon = 1/255$($\uparrow$) | Mean Accuracy CIFAR-10-C ($\uparrow$) |
|---|---|---|---|---|---|---|
| | CNN | $91.29 \pm 0.14$ | $50.49 \pm 0.62$ | $30.92 \pm 0.88$ | $10.78 \pm 0.33$ | 67.96 |
| | WD | $91.04 \pm 0.21$ | $48.40 \pm 0.36$ | $30.37 \pm 0.98$ | $10.72 \pm 0.91$ | 66.92 |
| | WD + Reg. ($\lambda = 0.01, h = 1, \rho_{diff} = 0$) | $89.03 \pm 0.27$ | $48.58 \pm 0.70$ | $39.77 \pm 1.02$ | $25.00 \pm 1.21$ | 73.85 |
| | WD + Reg. ($\lambda = 0.01, h = 1$) | $88.91 \pm 0.37$ | $48.93 \pm 0.42$ | $40.50 \pm 0.98$ | $25.94 \pm 1.37$ | 73.97 |
| | WD + Reg. ($\lambda = 0.01, h = 3, \rho_{diff} = 0$) | $89.86 \pm 0.33$ | $50.74 \pm 0.57$ | $41.60 \pm 0.99$ | $25.96 \pm 1.23$ | 73.60 |
| | WD + Reg. ($\lambda = 0.01, h = 3$) | $90.02 \pm 0.22$ | $50.38 \pm 0.37$ | $41.34 \pm 0.79$ | $25.86 \pm 1.07$ | 73.43 |
| | WD + Reg. ($\lambda = 0.05, h = 1$) | $89.05 \pm 0.27$ | $49.06 \pm 0.38$ | $40.83 \pm 0.70$ | $26.24 \pm 1.02$ | 73.76 |
| | WD + Reg. ($\lambda = 0.05, h = 3$) | $89.86 \pm 0.14$ | $50.85 \pm 0.71$ | $41.81 \pm 1.15$ | $26.79 \pm 1.52$ | 74.04 |
| | WD + Reg. ($\lambda = 0.1, h = 1, \rho_{diff} = 0$) | $88.91 \pm 0.10$ | $49.45 \pm 0.67$ | $41.27 \pm 1.28$ | $26.73 \pm 1.59$ | 73.66 |
| | WD + Reg. ($\lambda = 0.1, h = 1$) | $88.91 \pm 0.15$ | $49.16 \pm 0.95$ | $40.95 \pm 1.56$ | $26.39 \pm 1.84$ | 73.84 |
| | WD + Reg. ($\lambda = 0.1 h = 3, \rho_{diff} = 0$) | $89.95 \pm 0.14$ | $50.32 \pm 0.42$ | $41.11 \pm 0.52$ | $25.88 \pm 0.59$ | 74.04 |
| ResNet-20 | WD + Reg. ($\lambda = 0.1, h = 3$) | $89.85 \pm 0.12$ | $50.65 \pm 0.39$ | $41.57 \pm 0.40$ | $26.33 \pm 0.77$ | 73.85 |
| | SD | $\mathbf{91.36 \pm 0.23}$ | $50.83 \pm 0.49$ | $32.98 \pm 1.06$ | $11.97 \pm 0.92$ | 67.48 |
| | SD + Reg. ($\lambda = 0.01 h = 1, \rho_{diff} = 0$) | $89.14 \pm 0.10$ | $49.57 \pm 1.11$ | $40.90 \pm 1.85$ | $25.95 \pm 1.99$ | 73.80 |
| | SD + Reg. ($\lambda = 0.01, h = 1$) | $89.48 \pm 0.48$ | $50.59 \pm 1.19$ | $42.61 \pm 1.79$ | $28.22 \pm 1.64$ | 73.92 |
| | SD + Reg. ($\lambda = 0.01, h = 3, \rho_{diff} = 0$) | $90.51 \pm 0.36$ | $52.56 \pm 0.75$ | $43.22 \pm 0.83$ | $27.35 \pm 0.70$ | 73.46 |
| | SD + Reg. ($\lambda = 0.01, h = 3$) | $90.54 \pm 0.14$ | $\mathbf{53.12 \pm 0.34}$ | $\mathbf{44.42 \pm 0.63}$ | $\mathbf{29.14 \pm 1.15}$ | 74.14 |
| | SD + Reg. ($\lambda = 0.05, h = 1$) | $89.20 \pm 0.24$ | $49.59 \pm 0.74$ | $41.05 \pm 0.89$ | $26.28 \pm 1.04$ | 74.24 |
| | SD + Reg. ($\lambda = 0.05, h = 3$) | $90.34 \pm 0.14$ | $53.07 \pm 0.29$ | $44.04 \pm 0.56$ | $28.62 \pm 0.96$ | 73.98 |
| | SD + Reg. ($\lambda = 0.1, h = 1, \rho_{diff} = 0$) | $89.32 \pm 0.23$ | $49.38 \pm 0.91$ | $40.63 \pm 1.35$ | $25.73 \pm 1.89$ | 74.14 |
| | SD + Reg. ($\lambda = 0.1, h = 1$) | $89.29 \pm 0.25$ | $50.06 \pm 0.73$ | $41.90 \pm 1.03$ | $27.58 \pm 1.08$ | $\mathbf{74.29}$ |
| | SD + Reg. ($\lambda = 0.1, h = 3, \rho_{diff} = 0$) | $90.32 \pm 0.20$ | $52.22 \pm 0.48$ | $42.64 \pm 0.80$ | $27.11 \pm 0.97$ | 73.78 |
| | SD + Reg. ($\lambda = 0.1, h = 3$) | $90.28 \pm 0.14$ | $52.39 \pm 1.01$ | $43.29 \pm 1.16$ | $27.75 \pm 1.29$ | 74.07 |
| | CNN | $\mathbf{94.29 \pm 0.09}$ | $59.58 \pm 0.41$ | $53.04 \pm 0.59$ | $37.49 \pm 0.53$ | 73.38 |
| | WD | $93.73 \pm 0.06$ | $55.51 \pm 0.41$ | $49.84 \pm 0.33$ | $35.23 \pm 0.53$ | 72.87 |
| | SD | $93.97 \pm 0.15$ | $55.73 \pm 0.56$ | $50.29 \pm 0.55$ | $36.02 \pm 0.86$ | 73.48 |
| ResNet-9 | WD + Reg. ($\lambda = 0.01, h = 3$) | $93.18 \pm 0.19$ | $59.25 \pm 2.24$ | $56.08 \pm 3.22$ | $43.62 \pm 4.53$ | 76.41 |
| | WD + Reg. ($\lambda = 0.05, h = 3$) | $92.31 \pm 0.24$ | $56.82 \pm 1.16$ | $53.03 \pm 1.35$ | $40.33 \pm 1.52$ | 78.42 |
| | SD + Reg. ($\lambda = 0.01, h = 3$) | $93.09 \pm 0.30$ | $\mathbf{59.87 \pm 1.29}$ | $\mathbf{56.89 \pm 1.79}$ | $\mathbf{44.81 \pm 2.29}$ | 77.72 |
| | SD+ Reg. ($\lambda = 0.05, h = 3$) | $92.41 \pm 0.24$ | $57.05 \pm 1.02$ | $53.71 \pm 1.22$ | $44.79 \pm 7.77$ | $\mathbf{78.81}$ |
| | CNN | $90.38 \pm 0.10$ | $53.55 \pm 0.76$ | $54.05 \pm 1.44$ | $45.51 \pm 1.92$ | 68.09 |
| | WD | $\mathbf{90.51 \pm 0.34}$ | $49.87 \pm 1.59$ | $49.97 \pm 2.44$ | $40.76 \pm 3.27$ | 67.10 |
| | SD | $90.44 \pm 0.11$ | $51.04 \pm 0.89$ | $51.77 \pm 1.25$ | $43.39 \pm 1.63$ | 66.65 |
| EfficientNet-B0 | WD + Reg. ($\lambda = 0.01, h = 3$) | $88.97 \pm 0.31$ | $\mathbf{57.91 \pm 0.98}$ | $59.60 \pm 1.01$ | $53.37 \pm 1.19$ | 72.14 |
| | WD + Reg. ($\lambda = 0.05, h = 1$) | $88.14 \pm 0.42$ | $56.11 \pm 1.44$ | $57.78 \pm 1.62$ | $51.44 \pm 1.91$ | 73.98 |
| | WD + Reg. ($\lambda = 0.05, h = 3$) | $88.27 \pm 0.11$ | $56.78 \pm 0.69$ | $58.35 \pm 0.86$ | $52.04 \pm 1.13$ | 74.22 |
| | SD + Reg. ($\lambda = 0.01, h = 3$) | $89.18 \pm 0.52$ | $57.83 \pm 0.33$ | $\mathbf{59.68 \pm 0.44}$ | $\mathbf{53.50 \pm 0.59}$ | 71.87 |
| | SD+ Reg. ($\lambda = 0.05, h = 1$) | $88.18 \pm 0.46$ | $55.76 \pm 1.55$ | $57.23 \pm 1.64$ | $50.69 \pm 2.14$ | $\mathbf{74.45}$ |
| | SD + Reg. ($\lambda = 0.05, h = 3$) | $88.08 \pm 0.35$ | $56.72 \pm 1.86$ | $58.21 \pm 2.21$ | $51.91 \pm 2.61$ | 74.12 |

Table 10: Ablation Results (in % and averaged over 5 runs) of ResNet-20, ResNet-9, and EfficientNet-B0 trained with different convolution implementations and varying regularization on **CIFAR-100**, and evaluation against adversarial attacks, FGSM, PGD-40, and AutoAttack.

| | Variant | **Clean Val Acc.** in (%)(↑) | FGSM $\epsilon = 1/255$ (↑) | PGD-40 $\epsilon = 1/255$(↑) | AA $\epsilon = 1/255$ (↑) |
|---|---|---|---|---|---|
| ResNet-20 | CNN | **60.41 ± 0.33** | 14.36 ± 0.35 | 5.45 ± 0.30 | 1.17 ± 0.10 |
| | WD | 58.90 ± 0.19 | 12.84 ± 0.39 | 4.68 ± 0.16 | 1.01 ± 0.09 |
| | WD + Reg. ($\lambda = 0.01, h = 1, \rho_{diff} = 0$) | 55.31 ± 0.46 | 16.56 ± 0.49 | 12.65 ± 0.45 | 5.65 ± 0.38 |
| | WD + Reg. ($\lambda = 0.01, h = 1$) | 55.23 ± 0.26 | 16.75 ± 0.49 | 13.01 ± 0.60 | 5.93 ± 0.34 |
| | WD + Reg. ($\lambda = 0.01, h = 3, \rho_{diff} = 0$) | 56.50 ± 0.29 | 16.10 ± 0.72 | 11.63 ± 0.97 | 4.78 ± 0.58 |
| | WD + Reg. ($\lambda = 0.01, h = 3$) | 56.50 ± 0.27 | 16.01 ± 0.38 | 11.59 ± 0.42 | 4.96 ± 0.31 |
| | WD + Reg. ($\lambda = 0.05, h = 1$) | 55.55 ± 0.16 | 17.03 ± 0.46 | 13.26 ± 0.48 | 6.07 ± 0.46 |
| | WD + Reg. ($\lambda = 0.05, h = 3$) | 56.50 ± 0.29 | 16.60 ± 0.60 | 12.39 ± 0.56 | 5.28 ± 0.28 |
| | WD + Reg. ($\lambda = 0.1, h = 1, \rho_{diff} = 0$) | 55.41 ± 0.13 | 16.64 ± 0.66 | 12.71 ± 0.94 | 5.69 ± 0.72 |
| | WD + Reg. ($\lambda = 0.1, h = 1$) | 55.50 ± 0.47 | 17.03 ± 0.22 | 13.11 ± 0.62 | 5.91 ± 0.57 |
| | WD + Reg. ($\lambda = 0.1, h = 3, \rho_{diff} = 0$) | 56.77 ± 0.44 | 16.42 ± 0.48 | 12.26 ± 0.59 | 5.32 ± 0.35 |
| | WD + Reg. ($\lambda = 0.1, h = 3$) | 56.65 ± 0.39 | 16.85 ± 0.67 | 12.73 ± 0.65 | 5.59 ± 0.46 |
| | SD | 60.34 ± 0.44 | 13.87 ± 0.18 | 5.18 ± 0.09 | 1.13 ± 0.05 |
| | SD + Reg. ($\lambda = 0.01, h = 1, \rho_{diff} = 0$) | 56.32 ± 0.26 | 17.31 ± 0.57 | 13.26 ± 0.48 | 5.93 ± 0.39 |
| | SD + Reg. ($\lambda = 0.01, h = 1$) | 56.68 ± 0.50 | 17.40 ± 0.38 | 13.41 ± 0.60 | 6.03 ± 0.48 |
| | SD + Reg. ($\lambda = 0.01, h = 3, \rho_{diff} = 0$) | 58.27 ± 0.31 | 17.23 ± 0.23 | 12.47 ± 0.74 | 5.24 ± 0.53 |
| | SD + Reg. ($\lambda = 0.01, h = 3$) | 58.19 ± 0.23 | 17.20 ± 0.43 | 12.24 ± 0.51 | 5.11 ± 0.27 |
| | SD + Reg. ($\lambda = 0.05, h = 1$) | 56.71 ± 0.24 | 17.74 ± 0.33 | 13.73 ± 0.32 | 6.27 ± 0.32 |
| | SD + Reg. ($\lambda = 0.05, h = 3$) | 58.29 ± 0.57 | 17.24 ± 0.51 | 12.19 ± 0.39 | 4.95 ± 0.31 |
| | SD + Reg. ($\lambda = 0.1, h = 1, \rho_{diff} = 0$) | 56.56 ± 0.20 | 17.01 ± 0.34 | 12.91 ± 0.48 | 5.74 ± 0.26 |
| | SD + Reg. ($\lambda = 0.1, h = 1$) | 56.60 ± 0.37 | **17.59 ± 0.44** | **13.76 ± 0.56** | **6.40 ± 0.20** |
| | SD + Reg. ($\lambda = 0.1, h = 3, \rho_{diff} = 0$) | 58.32 ± 0.59 | 17.10 ± 0.45 | 12.46 ± 0.38 | 5.03 ± 0.32 |
| | SD + Reg. ($\lambda = 0.1, h = 3$) | 58.27 ± 0.41 | 17.00 ± 0.35 | 12.03 ± 0.60 | 5.01 ± 0.47 |
| ResNet-9 | CNN | 75.80 ± 0.28 | 30.41 ± 0.35 | 24.12 ± 0.27 | 10.13 ± 0.31 |
| | WD | 75.52 ± 0.34 | 29.97 ± 0.62 | 23.68 ± 0.48 | 10.80 ± 0.30 |
| | WD + Reg. ($\lambda = 0.01, h = 1$) | 73.85 ± 0.38 | 31.55 ± 0.71 | 28.00 ± 0.64 | 15.70 ± 0.37 |
| | WD + Reg. ($\lambda = 0.05, h = 1$) | 71.65 ± 0.29 | 28.28 ± 0.32 | 24.11 ± 0.31 | 12.84 ± 0.09 |
| | WD + Reg. ($\lambda = 0.05, h = 3$) | 72.69 ± 0.23 | 30.76 ± 0.20 | 26.72 ± 0.40 | 14.67 ± 0.39 |
| | WD + Reg. ($\lambda = 0.1, h = 1$) | 71.59 ± 0.28 | 28.67 ± 0.31 | 24.43 ± 0.40 | 13.04 ± 0.38 |
| | WD + Reg. ($\lambda = 0.1, h = 3$) | 72.74 ± 0.17 | 30.62 ± 0.19 | 26.52 ± 0.29 | 14.71 ± 0.36 |
| | SD | **76.06 ± 0.28** | 30.73 ± 0.16 | 24.65 ± 0.32 | 11.48 ± 0.37 |
| | SD + Reg. ($\lambda = 0.01, h = 1$) | 74.00 ± 0.27 | 31.97 ± 0.18 | 28.64 ± 0.37 | 16.42 ± 0.29 |
| | WD + Reg. ($\lambda = 0.01, h = 3$) | 74.75 ± 0.31 | 33.66 ± 0.18 | 30.31 ± 0.30 | 17.62 ± 0.37 |
| | SD + Reg. ($\lambda = 0.01, h = 3$) | 75.27 ± 0.27 | **34.04 ± 0.41** | **30.95 ± 0.35** | **18.26 ± 0.32** |
| | SD + Reg. ($\lambda = 0.05, h = 1$) | 71.70 ± 0.17 | 28.81 ± 0.43 | 24.75 ± 0.48 | 13.24 ± 0.43 |
| | SD + Reg. ($\lambda = 0.05, h = 3$) | 72.77 ± 0.23 | 31.07 ± 0.37 | 27.26 ± 0.46 | 15.15 ± 0.47 |
| | SD + Reg. ($\lambda = 0.1, h = 1$) | 71.56 ± 0.28 | 28.53 ± 0.33 | 24.41 ± 0.41 | 13.14 ± 0.35 |
| | SD + Reg. ($\lambda = 0.1, h = 3$) | 72.86 ± 0.24 | 31.07 ± 0.57 | 27.20 ± 0.64 | 15.08 ± 0.31 |
| EfficientNet-B0 | CNN | 62.09 ± 0.75 | 26.33 ± 1.07 | 24.99 ± 2.70 | 18.76 ± 3.37 |
| | WD | **63.49 ± 0.32** | 23.55 ± 0.53 | 18.13 ± 2.33 | 9.91 ± 2.46 |
| | WD + Reg. ($\lambda = 0.01, h = 1$) | 60.02 ± 0.24 | 26.00 ± 0.43 | 26.35 ± 0.58 | 20.97 ± 0.86 |
| | WD + Reg. ($\lambda = 0.01, h = 3$) | 60.11 ± 0.55 | 27.45 ± 0.88 | 27.69 ± 1.15 | 22.50 ± 1.31 |
| | WD + Reg. ($\lambda = 0.05, h = 1$) | 60.23 ± 0.50 | 25.45 ± 0.50 | 25.69 ± 0.54 | 20.34 ± 0.77 |
| | WD + Reg. ($\lambda = 0.1, h = 1$) | 60.16 ± 0.29 | 26.07 ± 0.62 | 26.35 ± 0.60 | 21.02 ± 0.66 |
| | WD + Reg. ($\lambda = 0.1, h = 3$) | 59.46 ± 0.07 | 27.27 ± 0.53 | 27.55 ± 0.57 | 22.57 ± 0.69 |
| | SD | 62.85 ± 0.29 | 25.11 ± 0.30 | 20.73 ± 1.95 | 13.21 ± 2.27 |
| | SD + Reg. ($\lambda = 0.01, h = 1$) | 61.24 ± 0.44 | 28.67 ± 0.62 | 28.70 ± 0.78 | 23.40 ± 1.19 |
| | WD + Reg. ($\lambda = 0.05, h = 3$) | 59.21 ± 0.42 | 27.44 ± 0.91 | 27.73 ± 1.03 | 22.93 ± 0.97 |
| | SD + Reg. ($\lambda = 0.01, h = 3$) | 60.49 ± 0.44 | **29.63 ± 0.61** | **30.24 ± 0.80** | **25.42 ± 1.21** |
| | SD + Reg. ($\lambda = 0.05, h = 1$) | 60.06 ± 0.41 | 27.79 ± 0.51 | 27.87 ± 0.53 | 22.80 ± 0.68 |
| | SD + Reg. ($\lambda = 0.05, h = 3$) | 59.65 ± 0.39 | 28.30 ± 0.65 | 28.82 ± 0.85 | 24.07 ± 1.08 |
| | SD + Reg. ($\lambda = 0.1, h = 1$) | 59.91 ± 0.27 | 27.55 ± 0.36 | 27.57 ± 0.39 | 22.44 ± 0.49 |
| | SD + Reg. ($\lambda = 0.1, h = 3$) | 59.72 ± 0.38 | 28.93 ± 0.77 | 29.36 ± 0.92 | 24.73 ± 1.14 |

## G  Adversarial Training on ImageNet

We show the results of our regularization on a ResNet-50 trained on ImageNet with adversarial training Table 11, based on the implementation in https://github.com/dedeswim/vits-robustness-torch. Training is executed against a single-step $L^\infty, \epsilon = 4/255$ PGD adversary. Our regularization improves both, clean and robust performance against non-regularized baselines. On ImageNet, we observe better results with WD for the same regularization hyperparameters.

Table 11: Performance of Adversarial Training of a ResNet-50 trained on ImageNet.

| Variant | Clean Val Acc. (↑) | Adversarial Acc. (↑) | |
|---|---|---|---|
| | | PGD $\epsilon = 4/255$ | AA $\epsilon = 4/255$ |
| CNN | 56.85 | 36.04 | 22.33 |
| SD | 56.50 | 35.96 | 18.66 |
| SD + Reg. ($\lambda = 0.05, h = 3$) | 57.61 | 35.85 | 22.25 |
| WD | 55.82 | 35.06 | 22.06 |
| WD + Reg. ($\lambda = 0.05, h = 3$) | **58.09** | **37.09** | **24.32** |
| WD + Reg. ($\lambda = 0.01, h = 3$) | 57.22 | 35.75 | 23.33 |
| WD + Reg. ($\lambda = 0.1, h = 3$) | 56.99 | 36.17 | 24.10 |
| WD + Reg. ($\lambda = 0.2, h = 3$) | 57.45 | 36.52 | 24.03 |
| WD + Reg. ($\lambda = 0.05, h = 1$) | 54.72 | 31.94 | 19.94 |

## H  Extended Ablation: Other regularization approaches

Table 12: Ablation on the **hierarchical regularization** approach on **EfficientNet-B0** trained with different convolution implementations on CIFAR-10 and evaluation against FGSM and PGD attacks. Results in % and averaged over 5 runs.

| Variant | Clean Val Acc. (↑) | Adversarial Acc. (↑) | |
|---|---|---|---|
| | | FGSM $\epsilon = 1/255$ | PGD-40 $\epsilon = 1/255$ |
| CNN | $90.38 \pm 0.10$ | $53.55 \pm 0.76$ | $54.05 \pm 1.44$ |
| WD | $\mathbf{90.51 \pm 0.34}$ | $49.87 \pm 1.59$ | $49.97 \pm 2.44$ |
| SD | $90.44 \pm 0.11$ | $51.04 \pm 0.89$ | $51.77 \pm 1.25$ |
| WD + Reg. ($\lambda = 0.05, h = 3$) | $88.27 \pm 0.11$ | $\mathbf{56.78 \pm 0.69}$ | $\mathbf{58.35 \pm 0.86}$ |
| WD + Reg. ($\lambda = 0.05, h = 3$, hierarchical) | $\underline{90.46 \pm 0.12}$ | $50.04 \pm 1.91$ | $50.22 \pm 2.82$ |
| SD + Reg. ($\lambda = 0.05, h = 3$) | $88.08 \pm 0.35$ | $\underline{56.72 \pm 1.86}$ | $\underline{58.21 \pm 2.21}$ |
| SD + Reg. ($\lambda = 0.05, h = 3$, hierarchical) | $90.36 \pm 0.43$ | $52.21 \pm 0.57$ | $53.13 \pm 0.54$ |

**Ablation.**  In order to further support the regularization approach, we included an ablation with a different regularization (Table 12). Here, instead of regularizing the coefficients belonging to the highest frequencies, we impose a penalty that enforces a hierarchy of frequencies based on magnitude starting with the highest magnitude on the lowest frequency. This regularization has a slightly positive impact on the robustness compared to its non-regularized counterpart but does not suppress higher frequencies. Therefore, it is intrinsically limited in improving robustness. We compare the regularization as described in Section 4.1 with a regularization approach, which ignores the first part of Equation 7 but rather forces the lowest frequencies to have a higher magnitude than subsequent frequencies for all frequencies, i.e. frequency $c_{0,0}$ has a higher frequency magnitude than $\{c_{0,1}, c_{1,0}, c_{1,1}\}$ and further $\{c_{0,1}, c_{1,0}, c_{1,1}\}$ has a higher magnitude than $\{c_{0,2}, c_{1,2}, c_{2,0}, c_{2,1}, c_{2,2}\}$ and so on. We denote this type of regularizaton with $\widetilde{\rho_{diff}}$. As we can see in Table 12

In this section, we additionally compare our proposed regularization with two other regularizations. First, to motivate the regularization of high frequencies, we regularize *inverse* to our original approach: instead of regularizing high frequencies, we regularize the low frequencies. In the case of kernel size $k = 3$ the penalty becomes:

$$P_{\text{inverse}}(C) = \|C_{:,:,0,0}\|_2 \cdot 2 + \max(\sum_{\substack{i,j=0 \\ (i=j)\neq 0}}^{1} (\|C_{:,:,i,j}\|_2) - \min(\|C_{:,:,3,3}\|_2), 0). \tag{9}$$

Second, we observe that some layers learn predominantly coefficients in the first row and column. To further investigate this observation, we regularize the lower quadrant, i.e. keeping the frequencies in the

Table 13: Ablation of **ResNet-20** with **different regularization types** on CIFAR-10 and evaluation against adversarial attacks, FGSM, PGD-40, AutoAttack, and corruption CIFAR-10-C. Reported is the mean accuracy in % and standard deviation over 5 runs.

| Variant | Clean Val Acc. ($\uparrow$) | FGSM $\epsilon = 1/255$ ($\uparrow$) | PGD-40 $\epsilon = 1/255$ ($\uparrow$) | AA $\epsilon = 1/255$ ($\uparrow$) | Mean Accuracy CIFAR-10-C ($\uparrow$) |
|---|---|---|---|---|---|
| CNN | $91.29 \pm 0.14$ | $50.49 \pm 0.62$ | $30.92 \pm 0.88$ | $10.78 \pm 0.33$ | 67.96 |
| WD | $91.04 \pm 0.21$ | $48.40 \pm 0.36$ | $30.37 \pm 0.98$ | $10.72 \pm 0.91$ | 66.92 |
| WD + Reg. ($\lambda = 0.01, h = 1$) | $88.91 \pm 0.37$ | $48.93 \pm 0.42$ | $40.50 \pm 0.98$ | $25.94 \pm 1.37$ | 73.97 |
| WD + Reg. ($\lambda = 0.01, h = 3$) | $90.02 \pm 0.22$ | $50.38 \pm 0.37$ | $41.34 \pm 0.79$ | $25.86 \pm 1.07$ | 73.43 |
| WD + Reg. ($\lambda = 0.05, h = 1$) | $89.05 \pm 0.27$ | $49.06 \pm 0.38$ | $40.83 \pm 0.70$ | $26.24 \pm 1.02$ | 73.76 |
| WD + Reg. ($\lambda = 0.05, h = 3$) | $89.86 \pm 0.14$ | $50.85 \pm 0.71$ | $41.81 \pm 1.15$ | $26.79 \pm 1.52$ | 74.04 |
| WD + Reg. ($\lambda = 0.1, h = 1$) | $88.91 \pm 0.15$ | $49.16 \pm 0.95$ | $40.95 \pm 1.56$ | $26.39 \pm 1.84$ | 73.84 |
| WD + Reg. ($\lambda = 0.1, h = 3$) | $89.85 \pm 0.12$ | $50.65 \pm 0.39$ | $41.57 \pm 0.40$ | $26.33 \pm 0.77$ | 73.85 |
| SD | $\mathbf{91.36 \pm 0.23}$ | $50.83 \pm 0.49$ | $32.98 \pm 1.06$ | $11.97 \pm 0.92$ | 67.48 |
| SD + Reg. ($\lambda = 0.01, h = 1$) | $89.48 \pm 0.48$ | $50.59 \pm 1.19$ | $42.61 \pm 1.79$ | $28.22 \pm 1.64$ | 73.92 |
| SD + Reg. ($\lambda = 0.01, h = 3$) | $90.54 \pm 0.14$ | $53.12 \pm 0.34$ | $\mathbf{44.42 \pm 0.63}$ | $\mathbf{29.14 \pm 1.15}$ | 74.14 |
| SD + Reg. ($\lambda = 0.05, h = 1$) | $89.20 \pm 0.24$ | $49.59 \pm 0.74$ | $41.05 \pm 0.89$ | $26.28 \pm 1.04$ | 74.24 |
| SD + Reg. ($\lambda = 0.05, h = 3$) | $90.34 \pm 0.14$ | $53.07 \pm 0.29$ | $44.04 \pm 0.56$ | $28.62 \pm 0.96$ | 73.98 |
| SD + Reg. ($\lambda = 0.1, h = 1$) | $89.29 \pm 0.25$ | $50.06 \pm 0.73$ | $41.90 \pm 1.03$ | $27.58 \pm 1.08$ | $\mathbf{74.29}$ |
| SD + Reg. ($\lambda = 0.1, h = 3$) | $90.28 \pm 0.14$ | $52.39 \pm 1.01$ | $43.29 \pm 1.16$ | $27.75 \pm 1.29$ | 74.07 |
| WD + Reg. ($\lambda = 0.01, h = 3$, inverse) | $90.45 \pm 0.22$ | $45.59 \pm 0.85$ | $23.85 \pm 1.13$ | $6.22 \pm 0.75$ | 65.41 |
| WD + Reg. ($\lambda = 0.05, h = 3$, inverse) | $90.47 \pm 0.14$ | $43.95 \pm 1.59$ | $20.39 \pm 0.87$ | $3.94 \pm 0.42$ | 64.49 |
| WD + Reg. ($\lambda = 0.1, h = 3$, inverse) | $90.62 \pm 0.23$ | $42.56 \pm 1.15$ | $18.69 \pm 2.10$ | $3.29 \pm 0.98$ | 64.57 |
| WD + Reg. ($\lambda = 0.01, h = 3$, quadrant) | $90.50 \pm 0.27$ | $51.76 \pm 0.87$ | $41.03 \pm 2.05$ | $24.37 \pm 2.96$ | 71.02 |
| WD + Reg. ($\lambda = 0.05, h = 3$, quadrant) | $90.08 \pm 0.20$ | $51.33 \pm 0.40$ | $40.20 \pm 1.10$ | $24.08 \pm 1.26$ | 71.55 |
| WD + Reg. ($\lambda = 0.1, h = 3$, quadrant) | $90.10 \pm 0.17$ | $51.14 \pm 0.42$ | $40.81 \pm 0.79$ | $24.79 \pm 1.05$ | 71.62 |
| SD + Reg. ($\lambda = 0.01, h = 3$, inverse) | $90.69 \pm 0.23$ | $47.23 \pm 0.58$ | $25.26 \pm 0.68$ | $6.36 \pm 0.57$ | 65.92 |
| SD + Reg. ($\lambda = 0.05, h = 3$, inverse) | $90.86 \pm 0.12$ | $45.94 \pm 1.35$ | $21.81 \pm 0.84$ | $4.36 \pm 0.55$ | 65.04 |
| SD + Reg. ($\lambda = 0.1, h = 3$, inverse) | $90.56 \pm 0.45$ | $44.10 \pm 0.97$ | $20.17 \pm 1.16$ | $3.66 \pm 0.42$ | 64.05 |
| SD + Reg. ($\lambda = 0.01, h = 3$, quadrant) | $90.67 \pm 0.06$ | $\mathbf{53.95 \pm 0.49}$ | $43.94 \pm 0.77$ | $27.23 \pm 1.19$ | 72.29 |
| SD + Reg. ($\lambda = 0.05, h = 3$, quadrant) | $90.49 \pm 0.11$ | $52.98 \pm 0.54$ | $42.31 \pm 1.21$ | $25.65 \pm 1.77$ | 72.22 |
| SD + Reg. ($\lambda = 0.1, h = 3$, quadrant) | $90.54 \pm 0.12$ | $53.02 \pm 0.66$ | $42.06 \pm 0.90$ | $25.77 \pm 1.26$ | 72.10 |

first column and first row ($c_{i,0}, c_{0,i}$, for $i \in \{0, \ldots, k\}$), with $k$ being the kernel size, *quadrant*. This way we do not regularize all coefficients corresponding to high frequencies. As a result, *quadrant* suppresses high frequencies on diagonal structures but allows vertical/horizontal high frequencies, while our original regularization enforces regularization independent of the orientation.

We present the results in Table 13 on ResNet-20. The *inverse* approach does not increase robustness, quite contrary it even decreases the accuracy under adversarial attacks and CIFAR-10-C, which strengthens the formulation of our original regularization. The *quadrant* regularization increases robustness but performs subpar compared to our original approach.

# I    Comparison-State-of-the-Art including Wang et al.

In the main paper, we compared our method to other robustness techniques such as FrequencyLowCut Pooling (Grabinski et al., 2022a) and Patch Gaussian Augmentation (Lopes et al., 2019) in Section 5.1. In Table 14 we additionally compare our proposed method to *Smoothing Kernel* (HFC) as proposed in Wang et al. (2020). However, in contrast to the results in Wang et al. (2020), we do not see favorable behavior of this method, but a decrease of clean validation accuracy and robust accuracy evaluated against PGD-40, FGSM, and AutoAttack. Since we can not replicate their results, we have reached out to the authors but have not received a reply. Furthermore, Table 14 also demonstrates, combining other robustness techniques with our proposed regularization even further increases the robustness.

Table 14 also provides the standard deviation over 5 runs to the tables Table 1 and Table 2 from the main paper.

Table 14: Comparison results (in % and averaged over 5 runs) of ResNet-20, ResNet-9, and EfficientNet-B0 against other robustness techniques (Grabinski et al., 2022a; Lopes et al., 2019; Wang et al., 2020) on **CIFAR-10**.We report the mean clean validation accuracy and robust accuracy against adversarial attacks: FGSM, PGD-40, and AutoAttack for $L^\infty, \epsilon = 1/255$, and the mean corruption accuracy on CIFAR-10-C. The proposed regularization improves over other robustness techniques on all models.

| | Variant | Clean Val Acc.(↑) | Robust Acc. $[L^\infty = \epsilon = 1/255]$ (↑) | | | Corruption (↑) |
|---|---|---|---|---|---|---|
| | | | FGSM | PGD-40 | AA | Mean Acc. |
| ResNet-20 | CNN | 91.29 ± 0.14 | 50.49 ± 0.62 | 30.92 ± 0.88 | 10.78 ± 0.33 | 67.96 |
| | CNN + FLC (Grabinski et al., 2022a) | **91.52 ± 0.29** | 52.49 ± 0.48 | 30.25 ± 0.42 | 8.48 ± 0.74 | 68.75 |
| | CNN + PaGA (Lopes et al., 2019) | 91.29 ± 0.15 | 50.36 ± 1.09 | 31.50 ± 1.82 | 11.38 ± 1.60 | 67.73 |
| | WD | 91.04 ± 0.21 | 48.40 ± 0.36 | 30.37 ± 0.98 | 10.72 ± 0.91 | 66.92 |
| | WD + FLC | 91.16 ± 0.17 | 51.40 ± 0.79 | 30.49 ± 1.36 | 8.08 ± 0.51 | 67.88 |
| | WD + PaGA | 91.01 ± 0.18 | 48.61 ± 0.53 | 31.44 ± 1.29 | 11.72 ± 1.45 | 67.87 |
| | SD | 91.36 ± 0.23 | 50.83 ± 0.49 | 32.98 ± 1.06 | 11.97 ± 0.92 | 67.48 |
| | SD + FLC | 91.42 ± 0.14 | 52.99 ± 0.43 | 33.21 ± 0.61 | 9.32 ± 0.29 | 68.44 |
| | SD + PaGA | 91.38 ± 0.27 | 50.68 ± 0.84 | 31.86 ± 1.20 | 10.91 ± 1.09 | 67.71 |
| | WD + Reg. | 89.86 ± 0.14 | 50.85 ± 0.71 | 41.81 ± 1.15 | 26.79 ± 1.52 | 74.04 |
| | WD + Reg + FLC | 89.65 ± 0.24 | 52.21 ± 0.48 | 42.02 ± 1.03 | 25.39 ± 1.69 | **75.28** |
| | WD + Reg + PaGA | 89.84 ± 0.15 | 50.09 ± 0.64 | 40.92 ± 1.24 | 25.79 ± 1.80 | 73.85 |
| | SD + Reg. | 90.54 ± 0.14 | 53.12 ± 0.34 | 44.42 ± 0.63 | **29.14 ± 1.15** | 74.14 |
| | SD + Reg + FLC | 90.37 ± 0.54 | **54.89 ± 1.81** | **45.13 ± 2.57** | 28.11 ± 3.56 | 74.41 |
| | SD + Reg + PaGA | 90.32 ± 0.15 | 52.47 ± 0.89 | 42.98 ± 1.33 | 27.30 ± 1.36 | 73.19 |
| ResNet-9 | CNN | 94.29 ± 0.09 | 59.58 ± 0.41 | 53.04 ± 0.59 | 37.49 ± 0.53 | 73.38 |
| | CNN + FLC (Grabinski et al., 2022a) | 94.24 ± 0.14 | 59.64 ± 0.36 | 53.47 ± 0.55 | 38.65 ± 0.73 | 73.81 |
| | CNN + PaGA (Lopes et al., 2019) | **94.33 ± 0.17** | 59.12 ± 0.55 | 52.62 ± 0.61 | 37.50 ± 0.48 | 73.72 |
| | CNN + HFC (Wang et al., 2020) | 87.62 ± 0.34 | 47.27 ± 0.87 | 42.41 ± 1.03 | 29.16 ± 1.08 | 32.85 |
| | WD | 93.73 ± 0.06 | 55.51 ± 0.41 | 49.84 ± 0.33 | 35.23 ± 0.53 | 72.87 |
| | WD + FLC | 93.73 ± 0.17 | 55.52 ± 0.23 | 50.07 ± 0.42 | 35.53 ± 0.25 | 73.00 |
| | WD + PaGA | 93.71 ± 0.07 | 55.62 ± 0.62 | 50.17 ± 0.64 | 35.50 ± 0.59 | 73.05 |
| | SD | 93.97 ± 0.15 | 55.73 ± 0.56 | 50.29 ± 0.55 | 36.02± 0.86 | 73.48 |
| | SD + FLC | 93.98 ± 0.10 | 56.07 ± 0.30 | 50.73 ± 0.44 | 36.36 ± 0.87 | 73.41 |
| | SD + PaGA | 93.79 ± 0.12 | 55.95 ± 0.46 | 50.73 ± 0.55 | 36.68 ± 0.75 | 73.32 |
| | WD + Reg. | 93.18 ± 0.19 | 59.25 ± 2.24 | 56.08 ± 3.22 | 43.62 ± 4.53 | 76.41 |
| | WD + Reg + FLC | 93.20 ± 0.19 | 59.64 ± 0.68 | 56.39 ± 0.84 | 44.24 ± 1.09 | 76.78 |
| | WD + Reg + PaGA | 92.15 ± 0.22 | 56.48 ± 0.80 | 52.64 ± 0.88 | 39.78 ± 1.23 | **78.14** |
| | SD + Reg. | 93.09 ± 0.30 | 59.87 ± 1.29 | 56.89 ± 1.79 | 44.81 ± 2.29 | 77.72 |
| | SD + Reg + FLC | 93.43 ± 0.35 | **60.80 ± 0.55** | **58.18 ± 0.69** | **46.08 ± 1.03** | 77.60 |
| | SD + Reg + PaGA | 93.14 ± 0.40 | 59.86 ± 1.21 | 57.24 ± 1.49 | 45.06 ± 1.62 | 77.54 |
| EfficientNet-B0 | CNN | 90.38 ± 0.10 | 53.55 ± 0.76 | 54.05 ± 1.44 | 45.51 ±1.92 | 68.09 |
| | CNN + FLC (Grabinski et al., 2022a) | 89.68 ± 0.26 | 51.92 ± 0.93 | 53.09 ± 0.98 | 45.37 ± 1.10 | 69.72 |
| | CNN + PaGA (Lopes et al., 2019) | 90.72 ± 0.33 | 54.18 ± 0.71 | 54.97 ± 1.29 | 46.64 ± 1.52 | 69.31 |
| | CNN + HFC (Wang et al., 2020) | 79.96 ± 1.30 | 42.40 ± 1.90 | 43.85 ± 2.03 | 37.34 ± 1.94 | 22.80 |
| | WD | 90.51 ± 0.34 | 49.87 ± 1.59 | 49.97 ± 2.44 | 40.76 ±3.27 | 67.10 |
| | WD + FLC | 89.60 ± 0.23 | 46.29 ± 1.30 | 46.66 ± 1.70 | 37.32 ± 2.21 | 68.54 |
| | WD + PaGA | 90.82 ± 0.21 | 48.77 ± 0.84 | 49.14 ± 1.80 | 39.50 ± 2.46 | 67.89 |
| | SD | 90.44 ± 0.11 | 51.04 ± 0.89 | 51.77 ± 1.25 | 43.39 ± 1.63 | 66.65 |
| | SD + FLC | 89.47 ± 0.15 | 47.91 ± 0.74 | 48.55 ± 1.22 | 39.92 ± 1.62 | 67.61 |
| | SD + PaGA | **90.91 ± 0.23** | 50.92 ± 0.63 | 51.74 ± 0.58 | 42.65 ± 0.75 | 68.13 |
| | WD + Reg. | 88.97 ± 0.31 | **57.91 ± 0.98** | 59.60 ± 1.01 | 53.37 ± 1.19 | 72.14 |
| | WD + Reg + FLC | 87.67 ± 0.32 | 52.68 ± 2.67 | 54.66 ± 2.83 | 48.14 ± 3.58 | 71.17 |
| | WD + Reg + PaGA | 88.49 ± 0.20 | 55.15 ± 0.77 | 56.70 ± 0.71 | 50.05 ± 0.85 | **74.16** |
| | SD + Reg. | 89.18 ± 0.52 | 57.83 ± 0.33 | **59.68 ± 0.44** | **53.50 ± 0.59** | 71.87 |
| | SD + Reg + FLC | 87.90 ± 0.32 | 54.15 ± 1.03 | 56.38 ± 0.97 | 50.31 ± 1.25 | 71.10 |
| | SD + Reg + PaGA | 89.66 ± 0.22 | 56.72 ± 0.79 | 58.56 ± 0.86 | 51.87 ± 1.13 | 72.39 |

## J   Perturbations in the frequency domain

In this section, we visualize common corruptions and FGSM attacks in frequency space. Therefore, we compute the average error in the FFT spectrum between the attack images and their clean counterparts.

**Common Corruptions.** Figure 10 shows an example image of the CIFAR-10-C test with various corruptions while Figure 11 shows the error of these corruptions in FFT space (as already shown in Yin et al. (2019)). Corruptions affect the frequency spectrum differently *e.g. pixelate* affects only high frequencies, *brightness* only low frequencies, and others like *Gaussian noise* affect the entire spectrum.

**FGSM.** Figure 12 shows the attack spectrum of an FGSM-attack with $\epsilon = 1/255, L^\infty$ on a ResNet-20, ResNet-9, and Efficientnet-B0 trained with CIFAR-10. We find perturbations for the models under the regular implementation of convolutions (non-robust), and regularized *weight/signal decomposition* layers. Regular convolutions are attacked in almost the entire spectrum, but primarily in the highest frequencies (top-left). Under regularization, we observe a shift of attacks to lower frequencies which indicates a successful

defense against high-frequency perturbations. We observe differences in the spectrum depending on the architecture.

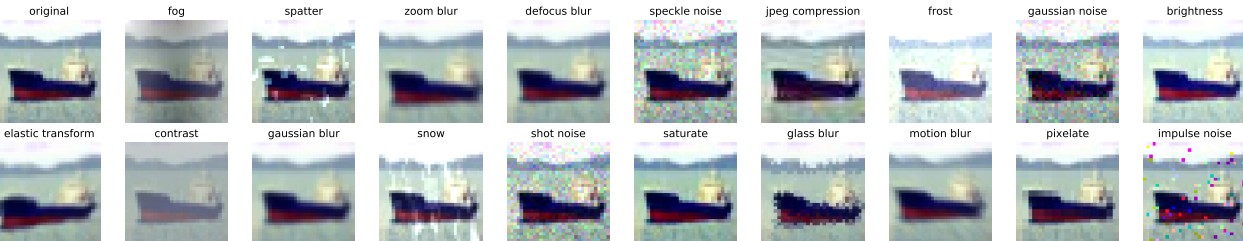

Figure 10: Visualizations of the corruptions in CIFAR-10-C.

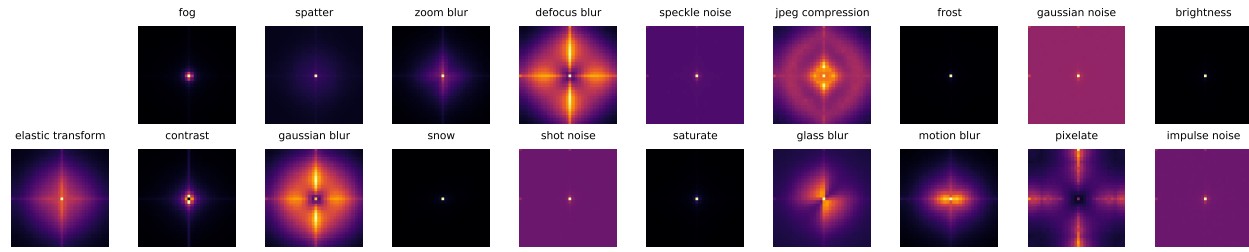

Figure 11: Mean absolute error in FFT spectrum of the corrupted CIFAR-10 test set (severity level 5) and clean test set. Corruptions affect the frequency spectrum differently *e.g. pixelate* affects only high frequencies, *brightness* only low frequencies, and others like *Gaussian noise* affect the entire spectrum. Black indicates no error, and bright colors indicate larger errors. The lowest frequencies are located in the center, high frequencies are located at the edges.

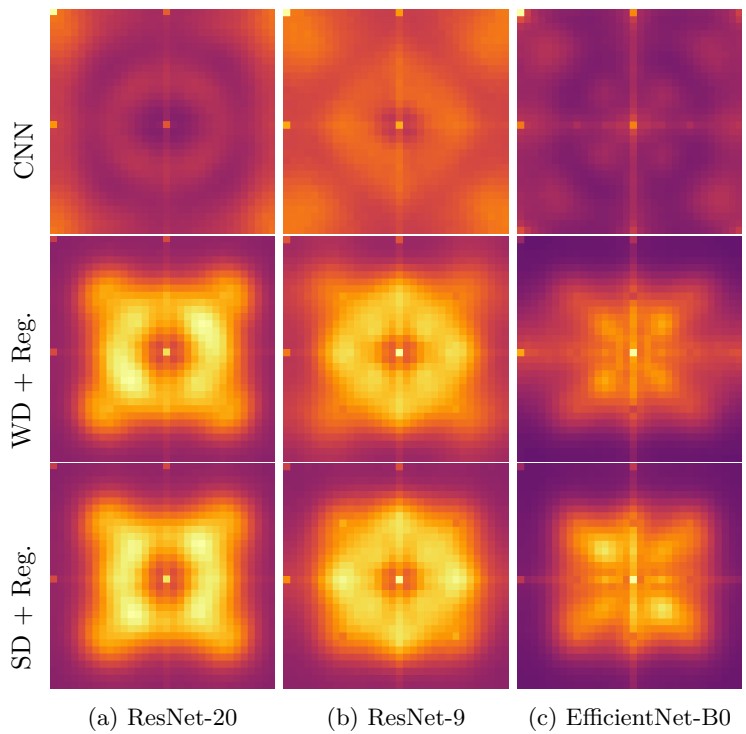

(a) ResNet-20      (b) ResNet-9      (c) EfficientNet-B0

Figure 12: Mean absolute error in the FFT spectrum of FGSM-attacks ($\epsilon = 1/255$, $L^\infty$) on CIFAR-10 models with different (regularized) convolution modifications and the clean test set. Regular convolutions are attacked in almost the entire spectrum, but primarily in the highest frequencies (top-left). After regularization, the adversarial attacks shift from high frequencies to lower ones. The architectures are attacked differently. Black indicates no error, and bright colors indicate larger errors. The lowest frequencies are located in the center, high frequencies are located at the edges.

## K   Details on shape bias

Accompanying Table 4, we also plot the texture-shape bias in Figure 13 using Geirhos et al. (2021) for our ImageNet (Deng et al., 2009) models, EfficinetNet-B0 and ConvNext-Tiny (Liu et al., 2022) for our decomposition approaches with and without regularization.

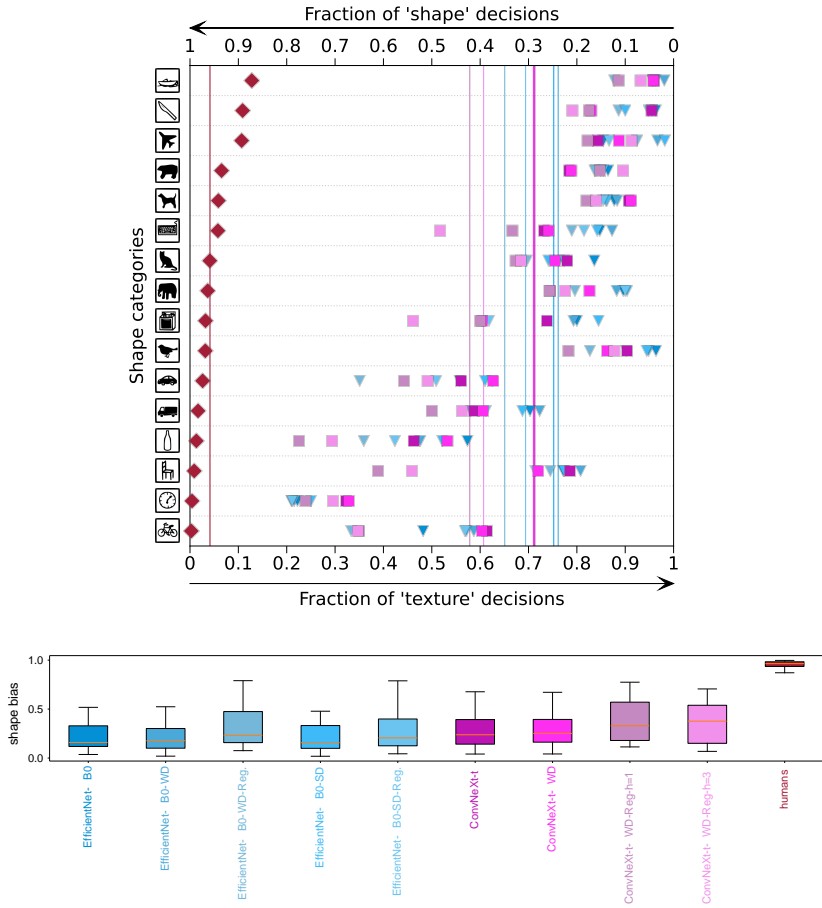

Figure 13: Top: Shape vs. texture bias: category-level plot. Bottom: Box-plot: shape vs. texture bias for all variants of EfficientNet-B0 and ConvNeXt-Tiny. Both plots were generated by Geirhos et al. (2021). High values indicate a shape bias, and low values a texture bias.

## L   Evolution Plots

In the following, we show further and more detailed evolution plots of the DCT-II coefficients in all convolution layers of ResNet-20 (Figure 14), ResNet-9 (Figure 15), and EfficientNet-B0 (Figure 16) between regular and adversarial training. The latter adversarial training is based on FGSM with an early stopping approach w.r.t the PGD-40 accuracy as presented in Wong et al. (2020).

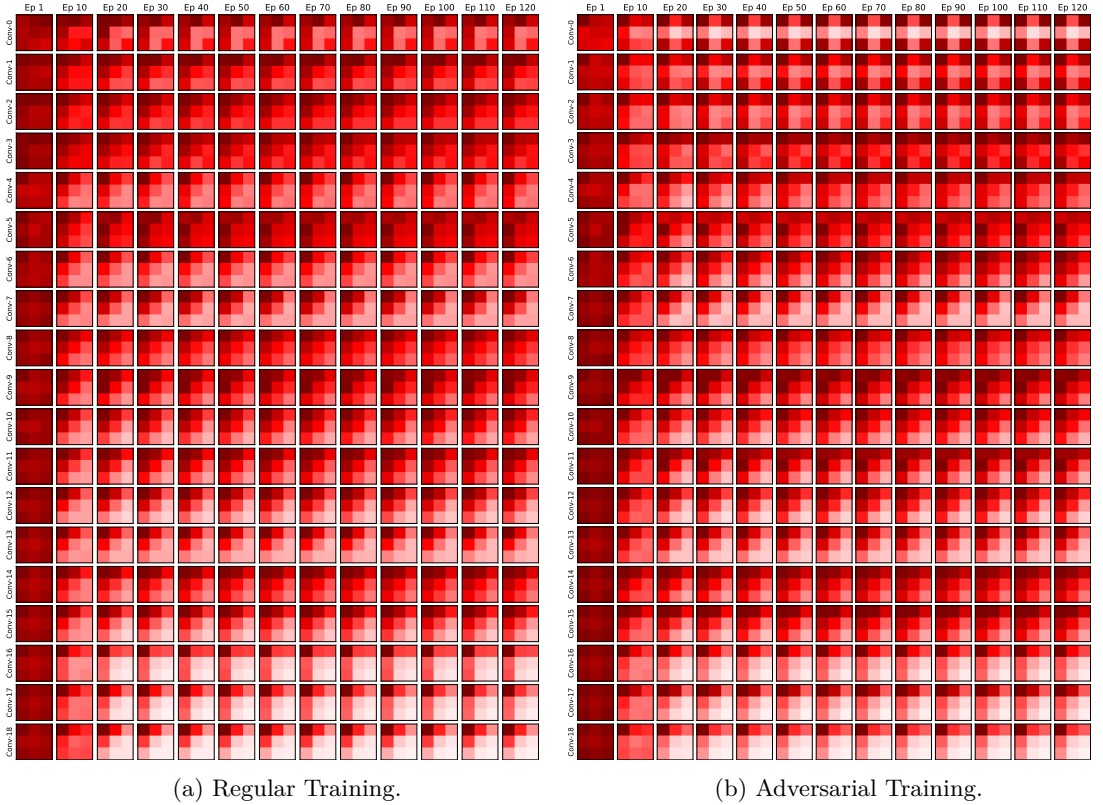

(a) Regular Training.

(b) Adversarial Training.

Figure 14: Evolution of the coefficients in a ResNet-20 trained on CIFAR-10.

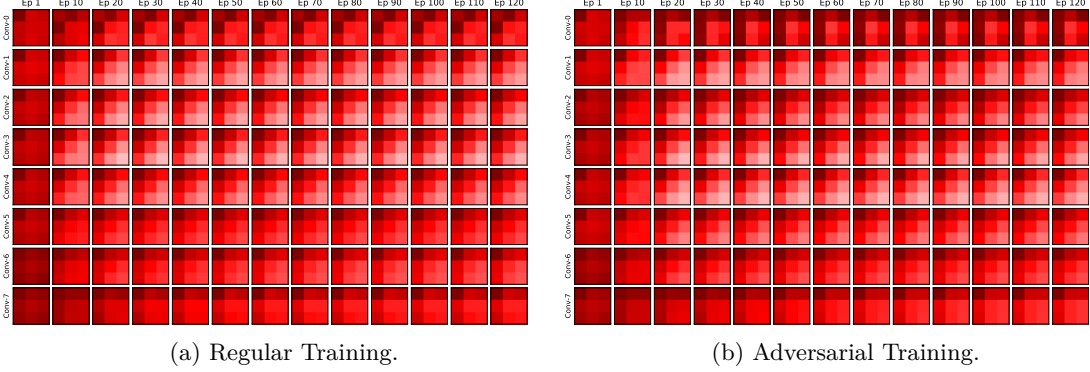

(a) Regular Training.

(b) Adversarial Training.

Figure 15: Evolution of the coefficients in a ResNet-9 trained on CIFAR-10.

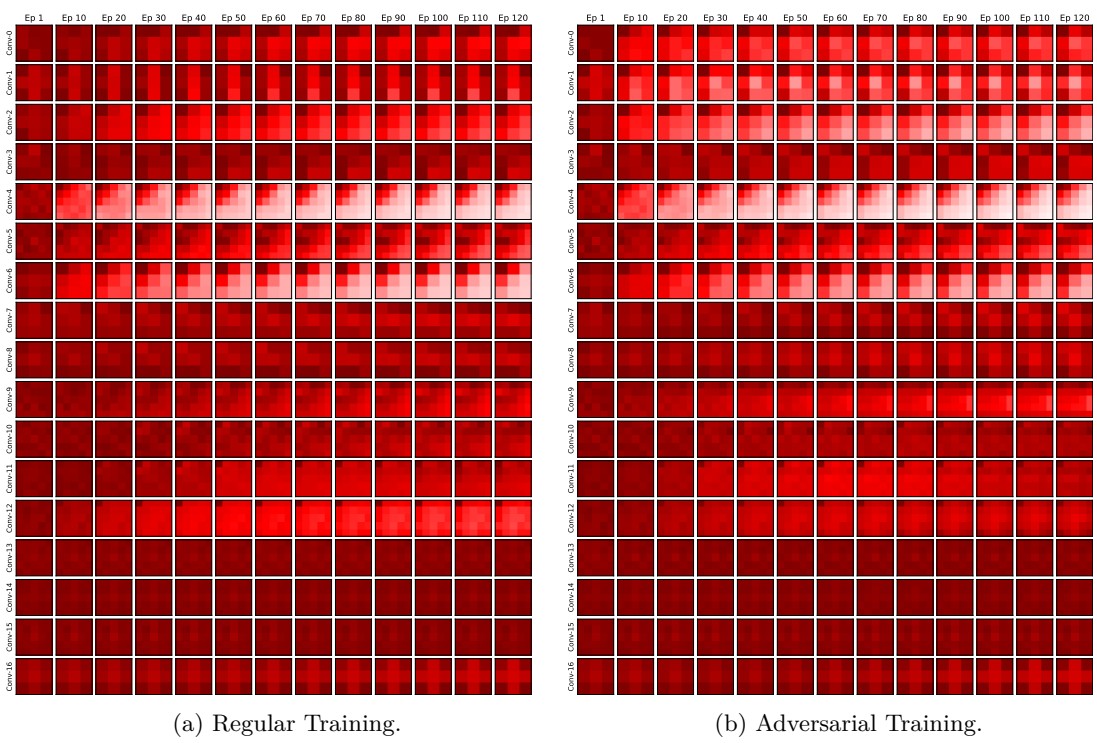

(a) Regular Training.

(b) Adversarial Training.

Figure 16: Evolution of the coefficients in a EfficientNet-B0 trained on CIFAR-10.

## M   Tables from Main Paper with Standard Deviation

In this section, we provide a more detailed version of Table 1 from the main paper. In Table 15 we also show the standard deviation over 5 runs.

Table 15: Results (in % and averaged over 5 runs) of ResNet-20 trained with different convolution implementations and regularization on **CIFAR-100, SVHN, TinyImageNet, MNIST**. We report the mean clean validation accuracy and robust accuracy against adversarial attacks: FGSM, PGD-40, and AutoAttack for $L^\infty, \epsilon = 1/255$. For MNIST we set $\epsilon = 16/255$. The usage of the proposed regularization improves the robustness on all datasets.

| Dataset | Model | Variant | Clean ($\uparrow$) Val Acc. | Robust Acc. $[L^\infty, \epsilon = \{1, 16^*\}/255]$ ($\uparrow$) FGSM | PGD-40 | AA |
|---------|-------|---------|------------|------|--------|-----|
| CIFAR-100 | ResNet-20 | CNN | **60.41 ± 0.33** | 14.36 ± 0.35 | 5.45 ± 0.30 | 1.17 ± 0.10 |
| | | WD | 58.90 ± 0.19 | 12.84 ± 0.39 | 4.68 ± 0.16 | 1.01 ± 0.09 |
| | | SD | 60.34 ± 0.44 | 13.87 ± 0.18 | 5.18 ± 0.09 | 1.13 ± 0.05 |
| | | WD + Reg. | 56.65 ± 0.39 | 16.85 ± 0.67 | **12.73 ± 0.65** | **5.59 ± 0.46** |
| | | SD + Reg. | 58.19 ± 0.23 | **17.20 ± 0.43** | 12.24 ± 0.51 | 5.11 ± 0.27 |
| SVHN | ResNet-20 | CNN | 96.31 ± 0.09 | 83.84 ± 0.65 | 79.94 ± 0.80 | 69.81 ± 1.12 |
| | | WD | **96.35 ± 0.15** | 83.52 ± 0.27 | 80.01 ± 0.41 | 71.25 ± 0.71 |
| | | SD | 96.34 ± 0.05 | 84.07 ± 0.27 | 80.64 ± 0.57 | 71.74 ± 0.85 |
| | | WD + Reg. | 96.28 ± 0.10 | 84.11 ± 0.27 | 81.21 ± 0.32 | **73.27 ± 0.64** |
| | | SD + Reg. | 96.34 ± 0.07 | **84.17 ± 0.19** | **81.23 ± 0.22** | 73.03 ± 0.84 |
| TinyImageNet | ResNet-9 | CNN | **53.20 ± 0.46** | 17.79 ± 0.14 | 16.76 ± 0.11 | 9.57 ± 0.24 |
| | | WD | 52.08 ± 0.26 | 17.11 ± 0.42 | 16.19 ± 0.43 | 9.40 ± 0.33 |
| | | SD | 52.12 ± 0.40 | 16.85 ± 0.28 | 15.88 ± 0.25 | 9.15 ± 0.22 |
| | | WD + Reg. | 51.25 ± 0.20 | 18.10 ± 0.28 | 17.34 ± 0.30 | 10.23 ± 0.28 |
| | | SD + Reg. | 51.22 ± 0.36 | **18.26 ± 0.36** | **17.40 ± 0.28** | **10.39 ± 0.20** |
| MNIST * | ResNet-20 | CNN | 99.68 ± 0.02 | 89.74 ± 5.63 | 45.37 ± 18.85 | 8.92 ± 8.33 |
| | | WD | 99.69 ± 0.05 | 90.45 ± 6.09 | 47.06 ± 13.02 | 10.22 ± 7.35 |
| | | SD | 99.65 ± 0.04 | **91.23 ± 2.87** | 54.08 ± 11.61 | 16.80 ± 11.47 |
| | | WD + Reg. | 99.69 ± 0.01 | 90.70 ± 3.92 | **55.84 ± 6.63** | **25.92 ± 3.25** |
| | | SD + Reg. | 99.69 ± 0.02 | 88.98 ± 3.35 | 50.02 ± 12.53 | 21.71 ± 8.26 |

## N   Qualitative Analysis of Attribution Maps

In the following section, we aim to understand how our regularization affects decisions by visualizing and interpreting attribution maps via SmoothGrad (Smilkov et al., 2017). We study the attributions of some selected ImageNet validation samples for the true label. We generate attributions from a regular ConvNeXt-Tiny, and the same network with WD and regularized WD layer as in Table 4. For all samples, we assert that all 3 models correctly predict the top-1 label. Our results in Figure 17 show that our proposed regularization (WD + Reg.) appears to shift attributions from edges and local texture shortcuts seen in unregularized networks (CNN or WD) to a more global understanding of the scene.

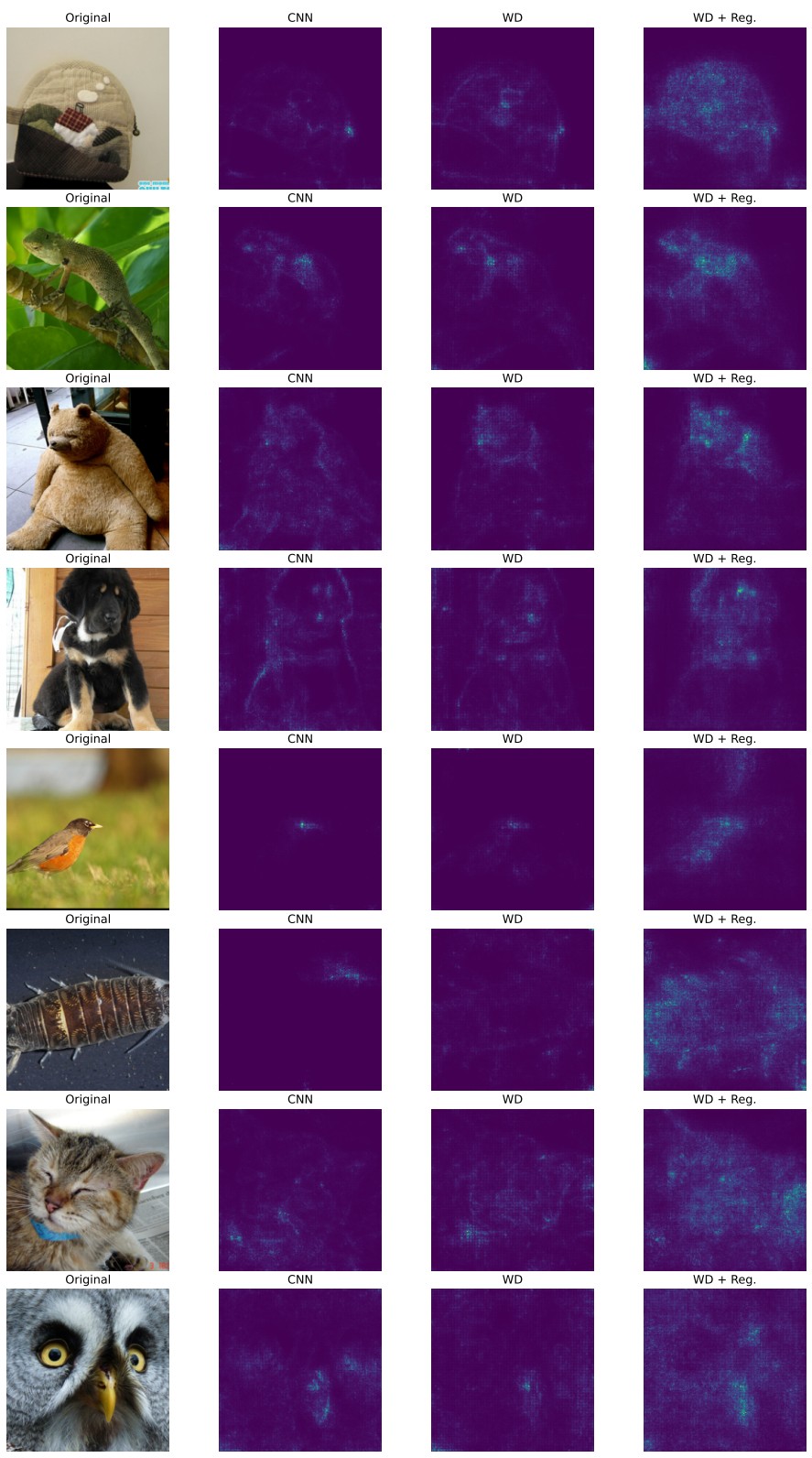

Figure 17: SmoothGrad attributions (Smilkov et al., 2017) of ConvNeXt-Tiny without changes to convolution layers (left), with unregularized WD (center), and regularized WD (right). The regularization appears to shift attributions from edges (first 4 rows) and local texture shortcuts (last 4 rows) to a more global understanding of the scene.

## O    Fourier Spectrum of Activations

Complementary to the results in Figure 6 we investigate the activations immediately after convolution layers of a ResNet-20 on a batch of clean CIFAR-10 test samples. However, this time we plot the centered magnitude of the Fourier magnitude (FFT). Note that the regularization only affects layers up to and including Bl.2.2. The results in Figure 18 show a shift towards lower frequencies in the affected layers of regularized models compared to non-regularized models. The shift is not as strong as for AT models. Figure 19 shows a couple of differences for better understanding. Note that in general, the shift is relatively small but still significantly affects robustness.

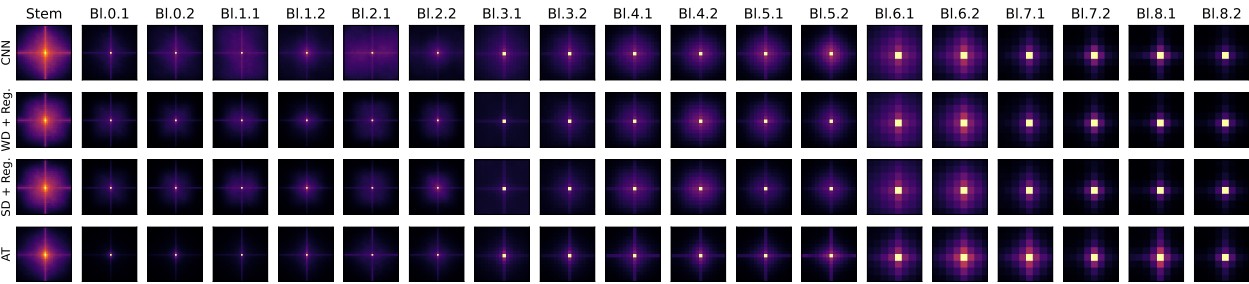

Figure 18: Visualization of the magnitude of the Fourier Spectrum of Activations in each convolution layer within a ResNet-20 trained on CIFAR-10.

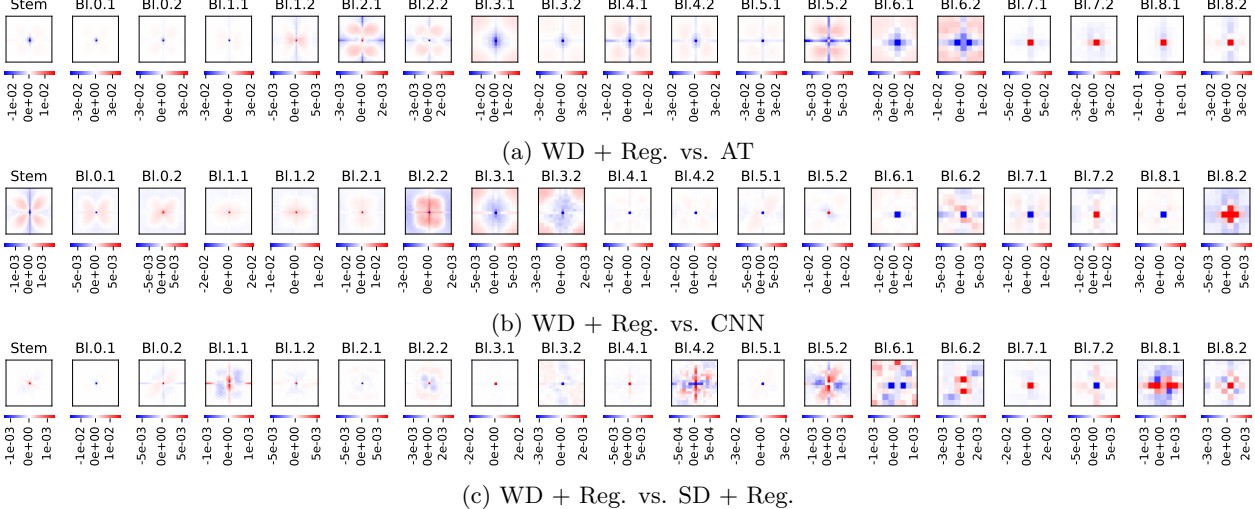

Figure 19: Comparison of Fourier domain activation magnitudes in a ResNet-20 trained on CIFAR-10, highlighting the distinctions between the WD + Reg. approach and various alternative methods

## P    Bridging Shape and Low-Frequency Bias

Our proposed regularization is based on the theoretical background of the relationship between shapes and frequencies. Geirhos et al. (2019) showed that neural networks tend to classify based on textures and not shapes which is not only different from human vision but also results in non-robustness due to the locality of the information. On the other hand, detection based on global information such as shapes must be more robust (Geirhos et al., 2021). Therefore, we want to induce a shape bias directly by our proposed regularization approach.

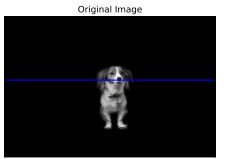 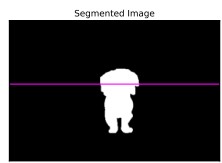 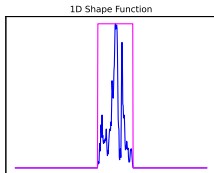 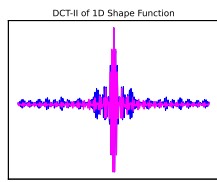 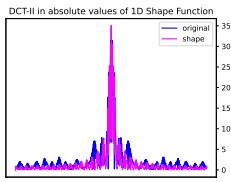

Figure 20: Shape and DCT-II spectrum for an object and its shape.

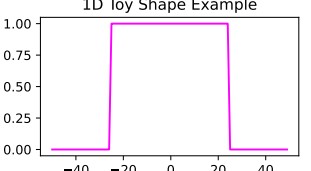 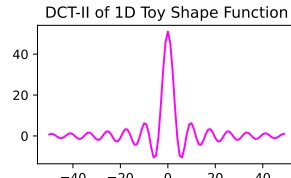

Figure 21: 1D shape toy example with its corresponding DCT-II basis transformation.

**Low-frequency components dominate shape signals.** For simplicity, let us consider a shape function in 1D, i.e. a rectangular pulse. For the Discrete Fourier Transform (DFT), it is well known that a rectangular pulse in the spatial domain directly corresponds to a sinc function, defined as $\mathrm{sinc}(\omega) = \frac{\sin(\omega)}{\omega}$, in the frequency domain and vice versa. The absolute value of this function assigns the highest values to the lowest frequency and the amplitude of the wave decreases as the frequency increases. The behavior of the DCT is related to that of the DFT and similar in practice.

Figure 20 visualizes the behavior for the DCT-II: Transforming the shape function of a 1D selected area of the image (Everingham et al.) into its frequency domain using the DCT-II basis results in a wave function whose amplitudes decrease as the frequency increases. We furthermore visualize this relationship on a toy example of a rectangle function and its transformation into the frequency domain (Figure 21). In the following, we provide the mathematical motivation of the resulting regularization term, which is given in Equation 7 in the main paper and which is inspired by this behavior. Specifically, we show the behavior of the DCT-II of a rectangular pulse.

Let $x \in C^N$ be a rectangular pulse of width $2M$ in the domain of symmetric, periodic signals of length N, approximating a square wave when $M = N/4$, i.e. $x(n) = 1$ for $-M \le n \le M$ and $x(n) = 0$ otherwise.

The DCT-II of $x$ is $X = \mathrm{DCT}(x)$, for $k = -\frac{N}{2}, \ldots, \frac{N}{2}$,

$$X(k) = \sum_{n=-M}^{M} \cos\left(\frac{\pi}{N}\left(n+\frac{1}{2}\right)k\right)$$

$$= \frac{1}{2}\sum_{n=-M}^{M} \exp\left(i\frac{\pi}{N}\left(n+\frac{1}{2}\right)k\right) + \exp\left(-i\frac{\pi}{N}\left(n+\frac{1}{2}\right)k\right) \tag{10}$$

$$= \frac{1}{2}\exp\left(i\frac{\pi}{2N}k\right)\sum_{n=-M}^{M}\left(\exp\left(i\frac{\pi}{N}k\right)\right)^n + \frac{1}{2}\exp\left(-i\frac{\pi}{2N}k\right)\sum_{n=-M}^{M}\left(\exp\left(-i\frac{\pi}{N}k\right)\right)^n$$

$$= \frac{1}{2}\exp\left(i\frac{\pi}{2N}k\right)\frac{\exp\left(i\frac{\pi}{N}k\right)^{-M} - \exp\left(i\frac{\pi}{N}k\right)^{M+1}}{1 - \exp\left(i\frac{\pi}{N}k\right)} \tag{11}$$

$$+ \frac{1}{2}\exp\left(-i\frac{\pi}{2N}k\right)\frac{\exp\left(-i\frac{\pi}{N}k\right)^{-M} - \exp\left(-i\frac{\pi}{N}k\right)^{M+1}}{1 - \exp\left(-i\frac{\pi}{N}k\right)}$$

$$= \frac{1}{2}\exp\left(i\frac{\pi}{2N}k\right)\frac{\exp\left(i\frac{\pi}{N}k\right)^{-M-\frac{1}{2}} - \exp\left(i\frac{\pi}{N}k\right)^{M+\frac{1}{2}}}{\exp\left(i\frac{\pi}{N}k\right)^{-\frac{1}{2}} - \exp\left(i\frac{\pi}{N}k\right)^{\frac{1}{2}}}$$

$$+ \frac{1}{2}\exp\left(-i\frac{\pi}{2N}k\right)\frac{\exp\left(i\frac{\pi}{N}k\right)^{M+\frac{1}{2}} - \exp\left(i\frac{\pi}{N}k\right)^{-M-\frac{1}{2}}}{\exp\left(i\frac{\pi}{N}k\right)^{\frac{1}{2}} - \exp\left(i\frac{\pi}{N}k\right)^{-\frac{1}{2}}} \tag{12}$$

$$= \frac{1}{2}\exp\left(i\frac{\pi}{2N}k\right)\frac{\exp\left(-i(M+\frac{1}{2})\frac{\pi}{N}k\right) - \exp\left(i(M+\frac{1}{2})\frac{\pi}{N}k\right)}{\exp\left(-i\frac{1}{2}\frac{\pi}{N}k\right) - \exp\left(i\frac{1}{2}\frac{\pi}{N}k\right)}$$

$$+ \frac{1}{2}\exp\left(-i\frac{\pi}{2N}k\right)\frac{\exp\left(i(M+\frac{1}{2})\frac{\pi}{N}k\right) - \exp\left(-i(M+\frac{1}{2})\frac{\pi}{N}k\right)}{\exp\left(i\frac{1}{2}\frac{\pi}{N}k\right) - \exp\left(-i\frac{1}{2}\frac{\pi}{N}k\right)}$$

$$= \frac{1}{2}\exp\left(i\frac{\pi}{2N}k\right)\frac{-2i\sin\left((M+\frac{1}{2})\frac{\pi}{N}k\right)}{-2i\sin\left(\frac{\pi}{2N}k\right)} + \frac{1}{2}\exp\left(-i\frac{\pi}{2N}k\right)\frac{2i\sin\left((M+\frac{1}{2})\frac{\pi}{N}k\right)}{2i\sin\left(\frac{\pi}{2N}k\right)} \tag{13}$$

$$= \frac{1}{2}\frac{\sin\left((M+\frac{1}{2})\frac{\pi}{N}k\right)}{\sin\left(\frac{\pi}{2N}k\right)}\left(\exp\left(i\frac{\pi}{2N}k\right) + \exp\left(-i\frac{\pi}{2N}k\right)\right) \tag{14}$$

$$\approx \frac{\sin\left(\frac{(M+\frac{1}{2})\pi}{N}k\right)}{\frac{\pi}{2N}k}\cos\left(\frac{\pi}{2N}k\right) = (2M+1)\frac{\sin\left((M+\frac{1}{2})\frac{\pi}{N}k\right)}{(M+\frac{1}{2})\frac{\pi}{N}k}\cos\left(\frac{\pi}{2N}k\right) \tag{15}$$

$$= (2M+1)\operatorname{sinc}\left(\left(M+\frac{1}{2}\right)\frac{\pi}{N}k\right)\cos\left(\frac{\pi}{2N}k\right)$$

In the second equality (10), we apply Euler's formula. Moving to (11) we apply the finite sums formula and cancel $\exp\left(i\frac{\pi}{N}k\right)^{\frac{1}{2}}$ and $\exp\left(i\frac{\pi}{N}k\right)^{-\frac{1}{2}}$ in line (12). Then, we again apply again Euler's formula in (13) and (14). The step to arrive at (15) is an approximation used for the DFT, following the reasoning that for small $\omega$, $\sin(\omega) = \omega$ and the term in the sin function in the denominator is small. As a result, the DCT of the rectangular pulse corresponds to a sinc function overlaid with a cosine. The amplitude of the resulting function decays as the frequency increases. For small $k$, the function approximates a sinc function. In the regularization term we propose in Equation 7, we, therefore, regularize the lower frequencies with the hierarchical term that encourages the maximal amplitudes of higher bands to be lower than the maximal frequency of lower bands, i.e. we regularize the $\max\left(\|\mathbf{C}_{[s],[d],L,M}\|_2 - \|\mathbf{C}_{[s],[d],0,0}\|_2\right)$ for $L, M = \{0, 1\}$. For high frequencies, we just encourage the amplitudes to be low in general, so that the full regularization is

$$\mathcal{R}(\mathbf{C}) = \|\mathbf{C}_{[s],[d],I,J}\|_2 \cdot I + \rho_{diff} \cdot \max\left(\|\mathbf{C}_{[s],[d],L,M}\|_2 - \|\mathbf{C}_{[s],[d],0,0}\|_2\right),$$

for $I, J = \{\lceil k/2\rceil, \ldots, k\}, L, M = \{0, 1\}$ and $[s] = \{0, \ldots, d_{\text{out}} - 1\}, [d] = \{0, \ldots, d_{\text{in}} - 1\}$.

