# OpenReview forum: "Improving Native CNN Robustness with Filter Frequency Regularization"
_TMLR — Accepted by TMLR_

### Review · Reviewer_KFrx · 2023-09-24

**Summary Of Contributions:**

This paper proposes a novel approach called filter frequency regularization to improve the native robustness of convolutional neural networks (CNNs) to various distribution shifts, including adversarial attacks, corruptions, and shape-biased datasets. The authors show that prioritizing low-frequency information in learned convolution weights leads to better model generalization and decision-making, as well as improved performance on adversarial attacks and out-of-distribution tests. They also demonstrate the effectiveness of their approach on several benchmark datasets and compare it to other state-of-the-art methods.

**Audience:**

Yes

**Broader Impact Concerns:**

No broader impact concerns

**Claims And Evidence:**

Yes

**Requested Changes:**

**Critical for acceptance**

- Clarifying the practical motivation for and definition of native robustness
- Revise related work to more clearly distinguish the contributions of this work from prior work
- In results tables, compare with adversarial training or a similarly strong method

**Would strengthen**

- The authors present a purely experimental paper, which demands very strong results; one possibility to strengthen the work without new experiments is to add theory exploring the relationship between frequency and robustness
- Some of the text in Figures 4 and 5 could be made larger to be more readable
- The authors mention that their method may be more aligned with human vision; showing this using some interpretability methods could be valuable

**Strengths And Weaknesses:**

**Strengths**

The specific method proposed by the authors is novel. Moreover, the robustness of their method to many types of image corruptions is a major strength. The method is also motivated by solid frequency analysis of existing networks in section 3, which is also a plus. The experiments are generally comprehensive and well-chosen. Finally, the writing and presentation of the paper are generally good, particularly the table format, figures, and notation.

**Weaknesses**

One weakness in my view is the motivation and explanation of native robustness. It's not clear to me what the practical value of native robustness is; if the goal is to produce networks that are robust to a variety of perturbations, one possibility is to simply adversarially train on all the possible perturbations. Moreover, there are existing works on training techniques robust to *unforeseen* adversarial attacks. The authors argue that, unlike adversarial training, their method does not require out-of-distribution training data. However, from another perspective, adversarial training does not require out-of-distribution training data either- instead, the process of generating adversarial examples for training can be viewed as part of the learning algorithm itself.

Another weakness is the similarity of this paper's perspective with other works. As the authors point out in section 2, there is literature already exploring the relationship between frequency and robustness, including works that propose using frequency as a tool to improve robustness. Moreover, the basis decomposition idea is not new either, as the authors note. Ideally, the distinctions between this work and prior work would be delineated more sharply.

Regarding the experiments, I think it is important in most of the tables to compare the proposed method with adversarial training or other strong methods of inducting robustness (even if these do not count as *natively* robust). I understand that the performance of these baselines would likely be expected to be better than the proposed method, but nevertheless, this would be important to have.

Also, in most cases, the improvement of the proposed method over the baseline of standard training appears relatively modest (although there are cases for which the improvements are quite large). Adding additional stronger baselines will make these improvements easier to evaluate.

Finally, as the authors note, the architecture itself impairs accuracy a little bit, which is not ideal, but overall this is still acceptable since the regularized performance is significantly stronger.

---

> ### Author Response · Authors · 2023-10-19
> **Response to Reviewer KFrx 1/2**
>
> We thank the reviewer for the insightful feedback. We will clarify each comment in the following:
>
> **W1/C1 (“Practical value of native robustness”):**
> We want to point out, that adversarial training on all possible perturbations as suggested by the reviewer is not feasible and also not predictable for 3 reasons:
> (1) Characterizing all possible perturbations is already very hard for specific problems and potentially impossible for vision in general (e.g., think about all the possible perturbations that can happen in autonomous driving). This problem hasn’t been solved to date.
> (2) Even if we were able to characterize all possible perturbations we could only train on a discrete set which would never reflect the full continuous space and thus risk overfitting
> (3) To effectively integrate this into current training approaches, we would have to significantly increase the training time to compensate for the new data.
>
> Contrarily, by native robustness we seek to add a low-frequency prior to the network to steer the training to a more generalizable representation without additional (perturbated/augmented) data. Still, we remain compatible with these methods and even adversarial training. In Sec. 5.3 we even show a reduced risk of robust overfitting after regularization.
>
> **W2/C2 (“Similarity with other work” and “Clearly distinguish”):**
> Indeed, the role of HF in generalization has been studied in literature before. But we significantly differ from previous works by the (combination) following aspects.
> (1) Regularization vs. bandpass filtering: previous methods often lowpass-filter signals which leads to a hard smoothing of the resulting feature maps and results in the active deletion of information that is often necessary to classify - especially in fine-grained classification problems. Instead, we only regularize the attenuation and thus effectively force the network to reweigh information without having to discard information. Our results in Tab. 2 indeed show that this is a superior way.
> (2) Data-independent and explicit attenuation: By regularizing weights we induce an explicit causal bias to the operator. E.g, an attenuation of feature maps would be implicit, but would highly depend on the frequency distribution of the inputs. Additionally, attenuating the filters results in a local suppression (i.e., in the patch) of HF, while a (global) feature-map regularization would affect the entire scene.
>
> **W3/C3 (“Comparison to AT in most tables missing”):** We apologize for the confusion. In fact, we already compare our proposed method with adversarial training on CIFAR-10. Due to our current organization, the results for Table 1/2 with AT are located in Table 5. As the trend is clear (improvement of adversarial robustness at a massive impairment of clean accuracy and robustness to other aspects of robustness) we did not perform adversarial training on other datasets. We have no reason to expect other trends, and sadly do not have the computational resources for extensive benchmarking. Does the reviewer have any concerns except completeness?
>
> **W4/C3 (“Adding stronger baselines”):** It is important to note, that while some improvements may be ‘modest’ our methods consistently improve robustness to multiple tests, whereas specialized techniques (e.g., AT) only improve certain aspects. Further, we have benchmarked related approaches. Can the reviewer give us a pointer to stronger baselines that are not computationally prohibitive?
>
> **W5 (“Accuracy impairment”):** We agree that this is not ideal and plan to pinpoint the exact cause and address it in future works.
>
> **C4 (“Theoretical explanation”):**
> Geirhos et al. (2019) showed that neural networks tend to classify based on textures and not shapes which is not only different from human vision but also results in non-robustness due to the locality of the information. On the other hand, detection based on global information such as shapes must be more robust (Geirhos et al., (2021)). In the following, we show the link between low-frequency prioritization and shape-biases and thus also to robustness.
>
> Shapes in 1D can be viewed as rectangles. The Fourier/Cosine transform of the rectangular function is given by the sinc function, which emphasizes low-frequencies with a hierarchical importance behavior from low frequencies to high frequencies. The same behavior is represented using our proposed regularization. Furthermore, applying the regularization on the convolution filters, allows a location-dependent frequency shift. Emphasizing the same behavior on feature maps, would however lead to a global low-frequency shift. See answer to W2/C2 for an additional discussion.
>
> Is the reviewer satisfied with this explanation (and W2/C2)? Do they suggest that we integrate this into the paper?

---

> > ### Comment · Reviewer_KFrx · 2023-10-20
> > **Thank you for your response.**
> >
> > I appreciate the changes made by the authors. The additional clarifications are also helpful, and it would be great to have these clarified in the paper as well. Given these changes, I believe this could potentially satisfy the critical acceptance criteria I listed in my review.

---

> ### Author Response · Authors · 2023-10-19
> **Response to Reviewer KFrx 2/2**
>
> **C5 (“Enlarge figure text”):** Done. We thank the reviewer for the suggestion! We changed the text size in Figure 4 and 5 for better readability.
>
> **C6 (“Interpretability methods”):** Thank you for the suggestion! We added the attributions of some selected ImageNet validation samples through SmoothGrad [1] in Appendix N. We generate attributions from a regular ConvNeXt-Tiny and the same network with WD and regularized WD layers. Our results show that our proposed regularization (WD + Reg.) appears to shift attributions from edges and local texture shortcuts seen in unregularized networks (CNN or WD) to a more human-like global understanding of the scene.
>
> [1] Daniel Smilkov, Nikhil Thorat, Been Kim, Fernanda B. Viégas, Martin Wattenberg:
> SmoothGrad: removing noise by adding noise. CoRR abs/1706.03825 (2017)

---

> ### Author Response · Authors · 2023-10-23
> **Update for Reviewer KFrx**
>
> Thank you very much for the first positive evaluation. According, to your suggestion we have now added the revision and added the following changes (including the changes of the last update):
>
> - We have included a further motivation and discussion on native robustness in Appendix A
> - We have included a thorough discussion about feature maps and weight regularization in Sec. 2
> - We have included the theoretical motivation of the proposed regularization term in Appendix P
> - We have enlarged the text in Fig. 4
> - We have added attribution maps in Appendix N
>
> Does this address your remaining concerns or do you have any other questions?

---

### Review · Reviewer_b5AR · 2023-09-24

**Summary Of Contributions:**

* This paper suggests introducing frequency regularization for DCT-II coefficients, based on the observation that adversarially trained models tend to focus more on low-frequency details in their initial layers. The application of this frequency regularization enhances the robustness of CNNs to both adversarial perturbations and common corruptions.
* This method for improving robustness does not require additional training data.
* The authors perform evaluations using modern CNN architectures like EfficientNet and ConvNext, assessing their effectiveness against both traditional attack methods like FGSM and PGD, as well as the most recent auto-attack.

**Audience:**

Yes

**Broader Impact Concerns:**

Since the study deals with adversarial attacks and defenses, it should include a statement addressing broader impacts statement

**Claims And Evidence:**

Yes

**Requested Changes:**

**Major:**
* This paper needs a summary section that consolidates all the results. Without the summary it is hard to appreciate the results.
* Compare against [1], [2], and [3]. The proposed method outperforms FLC, and prior research has already demonstrated FLC's superior robustness compared to [1], [2], and [3]. Nevertheless, it would be advantageous to include all these methods in a unified benchmark for future research purposes.
* Please see the weaknesses for any additional changes.

**Minor:**
* Please change references such as Tab. 1 to Table 1 for better readability.
* Kindly specify the relevant section in the appendix when making references in the main paper.

**Strengths And Weaknesses:**

**Strengths**:
(+) 1. The idea proposed by the author is interesting and worth investigating.
(+) 2. This paper conducts experiments across various datasets with different image sizes.
(+) 3. As shown in Table 1, the method improves both adversarial and corruption accuracy across all 3 architectures tested.
(+) 4. The results in Table 2 indicate that the proposed method is more robust than the previous Frequencylowcut pooling (FLC) and Patch gaussian augmentation (Paga).

**Weaknesses**:
(-) 1. The high-frequency regularization approach is intended to emulate the behavior seen in adversarial training. When analyzing the spectrum of feature maps, in Section 5.1, how do the layers subjected to regularization compare to those in a CNN baseline trained using adversarial methods?
(-) 2. Missing citation. Similar assumptions regarding the types of features learned by adversarially trained models have been made in a closely related study [4]. Please cite this study and discuss the two approaches in the related work section.
(-) 3. Important missing citations. Previous works have examined anti-aliasing to improve robustness, with a particular focus on mitigating aliasing in the downsampling operation in [1], [2], and [3]. However, this paper does not discuss or compare against these methods. It would be important to benchmark the proposed method against these methods.
(-) 4. In terms of training, how does the performance of the proposed method compare to FLC pooling, and also others? The discussion of the training time comparison will be essential for determining the practical feasibility of employing this method in downstream computer vision tasks.

[1] Richard Zhang. Making convolutional networks shift-invariant again. In International Conference on Machine Learning (ICML), pp. 7324-7334. PMLR, 2019.
[2] Xueyan Zou, Fanyi Xiao, Zhiding Yu, and Yong Jae Lee. Delving deeper into anti-aliasing in convnets. In British Machine Vision Conference (BMVC), 2020.
[3] Qiufu Li, Linlin Shen, Sheng Guo, and Zhihui Lai. Wavelet Integrated CNNs for Noise-Robust Image Classification. In Proceedings of the IEEE/CVF Conference on Computer Vision and Pattern Recognition (CVPR), 2020, pp. 7245-7254.
[4] Frequency Regularization for Improving Adversarial Robustness. In Practical-DL Workshop, Association for the Advancement of Artificial Intelligence (AAAI), 2023.

---

> ### Author Response · Authors · 2023-10-19
> **Response to Reviewer b5AR**
>
> We thank the reviewer for the insightful feedback. We will clarify each comment in the following:
>
> **W1 (“Spectrum of feature maps analysis”):** We included in Section O in the appendix (Figure 18) additional investigations of the spectrum of feature maps. This visualization shows that adversarially trained networks are shifted toward lower frequencies in comparison to clean-trained networks. Furthermore, as expected, our regularization also shows a shift towards lower frequencies, yet, not as intense as AT.
>
> **W2 (“Missing citation”):** We included the mentioned study as follows in our related work section:
> >Huang et al. (2023) propose a frequency regularization in the context of adversarial training, which contrarily to the previously listed works amplifies high frequencies, resulting in higher robust accuracy, at an impairment of clean accuracy.
>
> **W3/C2 (“Compare against [1], [2], and [3]”):** We will include additional comparisons using ResNet-20 on CIFAR-10 in a separate response in the next days.
>
> **W4 (“Training time comparison”):** We will include additional training time comparisons in a separate response in the next days.
>
> **C1 (“Summary Section”):** We already have summarized our contributions and results at the end of our introduction at the bottom of page 2. Could the reviewer clarify the requested changes or additions in more detail?
>
> **C4 (“Change tab. references”):** Done. We changed the references for better readability.
>
> **C5 (“References to the appendix”):** Done. We included references to the appendix in the main paper.
>
> **Broader impact statement:**
> Done. We thank the reviewer for bringing this to our attention. We have now included the following additional section in the appendix that addresses broader impacts.
>
> >Arms race: Adversarial training, which is a common technique to enhance model robustness, involves introducing worst-case perturbations into the training data. However, the very same techniques used for adversarial training can also be employed by malicious actors to craft adversarial attacks. As AI models become more robust to conventional adversarial attacks, adversaries may develop more sophisticated and potent attack strategies. This arms race in adversarial techniques could potentially lead to an escalation of cyber threats, with negative consequences for cybersecurity and data privacy.
>
> >Ethical and bias: While the paper's approach to prioritizing low-frequency information in decision-making can mitigate certain types of bias, it is crucial to recognize that bias is generally a multifaceted issue. Shifting the focus toward low-frequency information may introduce its own set of biases. For example, in computer vision, prioritizing shapes might lead to under representation or misrepresentation of fine-grained details, which can have implications in fields like medical diagnostics and object recognition. It is essential to carefully consider and address these potential biases to ensure fair and equitable AI systems.

---

> > ### Author Response · Authors · 2023-10-23
> > **Update to Reviewer b5AR**
> >
> > Thank you for the initial feedback! Accordingly, we have made the following updates to our submission:
> >
> > - We have added Fourier spectra of activations in Appendix O
> > - We have added the missing citation (Huang et al., (2023)) in Sec. 2
> > - We have added an additional anti-aliasing comparison in Tab. 2
> > - We have added computation time comparison in Tab. 3
> > - We have removed abbreviations in hyperrefs
> > - We have added hyperrefs to appendix sections
> > - We have added a broader impact section in Appendix A
> >
> > Do these changes in addition to the previous answers address your concerns? Do you have any further questions?

---

> > > ### Comment · Reviewer_b5AR · 2023-10-23
> > > **Thank you for the response**
> > >
> > > I thank the authors for their response. Their response has sufficiently addressed all my concerns and I have no further questions at this point. I am looking forward to the discussion with the other reviewers.

---

### Review · Reviewer_TkKe · 2023-10-10

**Summary Of Contributions:**

The authors observe that the cleanly-trained models often prioritize high-frequency information, whereas adversarial training enforces models to shift the focus to low-frequency details during training. Based on this observation, the authors propose a high-frequency penalization term in loss function to improve robustness to adversarial attacks. The authors conduct experiments to illustrate their findings and the effectiveness of their proposed method.

**Audience:**

Yes

**Broader Impact Concerns:**

No concern on ethical implications.

**Claims And Evidence:**

No

**Requested Changes:**

Please see "Strengths And Weaknesses."

I suggest the major revision.

**Strengths And Weaknesses:**

[Strengths]
The proposed method effectively improves the networks’ native robustness without training on adversarial samples.

[ Weaknesses & Suggestions to Authors]
1.	If the authors want to claim the relationship between the network’s robustness and the frequency spectrums of the convolutional kernels, there are at least three key problems the authors need to clarify:
(1) It is generally believed that the adversarial attack on the input image mainly affects the high frequencies of the input image. However, is there any evidence to show the adversarial attack mainly affects the high frequencies or low frequencies of the intermediate-layer feature? Spectrums of input images are not equivalent to spectrums of feature maps.
(2) What’s the relationship between the network’s robustness and the intermediate-layer feature spectrums?
(3) What’s the relationship between the intermediate-layer feature spectrums and the frequency spectrums of the convolution kernels?
It would be better if the authors theoretically analyze, or discuss, or conduct some experiments towards the above three concerns.

2.	In Equation (8) of this paper, the second term of the function R(C) seems to be ad-hoc.

3.	In a CNN, most kernels are of the size 3*3. Based on Equation (8), it’s hard to tell which frequencies are penalized. Is it meaningful to distinguish low frequencies and high frequencies from such a small 3*3 map?

4.	The size of ResNet-9 is too small. We suggest the authors to conduct experiments on more large SOTA models, such as ResNet-50, ResNet-101, ….

[ Minors]
1.	In Equation (8) of this paper, the authors use an informal index representation, such as “:, :, [k/2]:, [k/2]:”

The overall rating is major revision.

---

> ### Author Response · Authors · 2023-10-19
> **Response to Reviewer TkKe**
>
> We thank the reviewer for the insightful feedback. We will clarify each comment in the following:
>
> **W1.1 (“Any evidence to show the adversarial attack mainly affects the high frequencies or low frequencies of the intermediate-layer feature.“):**
> We do not have direct evidence for that but we noticed that AT shifts activations (and weights) to low-frequency information*. Our scope of this work is to mimic a (good) part of AT but to generalize to a broader set of robustness aspects - as such, analyzing the activations of adversarial attacks would not necessarily generalize to other kinds of robustness tests.
>
> \* We have included additional feature spectrum analyses in Section O in the appendix (Figures 18, 19). This visualization shows a shift towards lower frequencies for adversarially trained networks, especially in the first layers, which still holds for intermediate layers, which is not the case for the non-regularized clean-trained CNN. Similarly to AT, our proposed regularization also shifts the spectrum to lower frequencies.
>
> **W1.2/3 (“Relationship between the network’s robustness and the intermediate-layer feature spectrums, Relationship between the intermediate-layer feature spectrums and the frequency spectrums of the convolution kernels”):**
> The convolution theorem states that a convolution of an image with a kernel in a spatial domain is equivalent to the element-wise multiplication of an image with the kernel frequency domain. Therefore, roughly, if the convolutional kernel is attenuated in the frequency domain (e.g., by setting weights to 0), it will weigh the corresponding frequency band in the feature-map accordingly.
>
> Previous work has already shown the link between low-frequency bias and robustness (e.g., (Wang et al., 2020)).
>
> **W2 (“Second term of Equation (8) seems ad-hoc”):**
> Geirhos et al. (2019) showed that neural networks tend to classify based on textures and not shapes which is not only different from human vision but also results in non-robustness due to the locality of the information. On the other hand, detection based on global information such as shapes must be more robust (Geirhos et al., (2021)). In the following, we show the link between low-frequency prioritization and shape-biases resulting in more robustness and thus the motivation for our regularization.
>
> Shapes in 1D can be viewed as rectangles. The Fourier/Cosine transform of the rectangular function is given by the sinc function, which emphasizes low-frequencies with a hierarchical importance behavior from low frequencies to high frequencies. The same behavior is represented using our proposed regularization. Furthermore, applying the regularization on the convolution filters, allows a local, location-dependent frequency dependency. Emphasizing the same behavior on feature maps, would however lead to a global low-frequency shift. See answer to Reviewer KFrx W2/C2 for additional details.
>
> **W 3.1 (“It’s hard to tell which frequencies are penalized”):**
> Since we not only penalize but also establish a hierarchy in the magnitude of the coefficients the equation might not be straightforward to understand. This is why we visualize the regularization area in Fig. 5 and describe the approach in Sec. 4.1 - we have now updated Fig.5 to show the regularization of a 3x3 kernel. We establish a hierarchy between the lowest and second lowest coefficients and additionally regularize the coefficients corresponding to the highest frequencies in the convolution kernel.
>
> **W 3.2 (“Is it meaningful to distinguish low frequencies and high frequencies from such a small 3x3 map”):**
> Indeed, the granularity of the frequency regularization is rather small. Nonetheless, even small kernels can also be decomposed into frequencies. Overall our results as well as the visualization in Fig. 6 (and additional Fig. 18/19 in the updated appendix) show that this is still enough to shift activations into low-frequency parts. In Fig. 12 in the appendix, we also show the same effect on the attack spectrum of adversarial attacks.
>
> **W4 (“We suggest the authors to conduct experiments on more large SOTA models”):**
> We only use this network on low-resolution datasets, where too much downsampling (as in ResNet-50/101) is not reasonable. Furthermore, we show additional results on ResNet-20, and EfficientNet-B0. On ImageNet we do not use ResNet-9, but EfficientNet-B0 (17 convolution layers), ConvNeXt-tiny (22 convolution layers), and ResNet-50 for AT.
>
> The more recent ConvNeXt-tiny in fact already outperforms ResNet-50 on ImageNet and performs approximately on par with ResNet-101. That is why we decided to evaluate the more recent ConvNext-Network. We don’t see any additional value in testing more architectures. What would the benefit of additional experiments on other architectures be?
>
> **W5 (“Equation (8) of this paper, the authors use an informal index representation”):** Done. We have updated the notation.

---

> > ### Author Response · Authors · 2023-10-23
> > **Update to Reviewer TkKe**
> >
> > Thank you for the initial feedback! Accordingly, we have made the following updates to our submission:
> >
> > - We have added and analyzed Fourier spectra of activations in Appendix O in order to further motivate the regularization of convolution weights corresponding to feature maps in the first network layers
> > - We have included the theoretical motivation of the proposed regularization term in Appendix P
> > - We have changed Fig. 5 to show a 3x3 kernel
> > - We have written Eq. 8 in proper notation and improved the legibility
> > - We have updated indexing in Eq. 3.
> >
> > Do these changes in addition to the previous answers address your concerns? Do you have any further questions?

---

### Author Response · Authors · 2023-10-23
**Thanks to all reviewers and Summary of Changes**

We thank all reviewers for their feedback, which allowed us to improve our manuscript. In the following, we provide a summary of all changes we made in the submission. We hope our changes address all remaining concerns.

**Changes:**
- We have included a further motivation and discussion on native robustness in Appendix A (as suggested by reviewer KFrx in W1/C1)
- We have included a thorough discussion about feature maps and weight regularization in Sec. 2 (as suggested by KFrx W2/C2)
- We have included the theoretical motivation of the proposed regularization term in Appendix P (as suggested by reviewers KFrx in C4 & and TkKe in W2)
- We have enlarged the text in Fig. 4 (as suggested by KFrx in C5)
- We have added attribution maps in Appendix N (as suggested by reviewer KFrx in C6)
- We have added Fourier spectra of activations in Appendix O (as suggested by reviewers b5AR in W1 & TkKe in W1.1)
- We have added the missing citation (Huang et al., (2023)) in Sec. 2 (as suggested by b5AR in W2)
- We have added an additional anti-aliasing comparison in Tab. 2 (as suggested by b5AR in W3/C2)
- We have added computation time comparison in Tab. 3 (as suggested by b5AR in W4)
- We have removed abbreviations in hyperrefs (as suggested by reviewer b5AR in C4)
- We have added hyperrefs to appendix sections (as suggested in C5 by Reviewer b5AR)
- We have added a broader impact section in Appendix A (as suggested by b5AR "Broader impact statement")
- We have added and analyzed Fourier spectra of activations in Appendix O (as suggested by reviewer TkKe in W1, to motivate the regularization of the weights in deeper layers)
- We have changed Fig. 5 to show a 3x3 kernel (as suggested by TkKe W3.1)
- We have written Eq. 8 in proper notation and improved the legibility (as suggested by reviewer TkKe in W5)
- We have updated indexing in Eq. 3.

Changes are highlighted in blue in the updated PDFs.

---

### Public Comment · ~Kiran_Chari1 · 2024-01-13
**Missing discussion of relevant related work**

Congratulations on the publication of this work. A closely related work that is not discussed in the paper is "Fourier Sensitivity and Regularization of Computer Vision Models" (TMLR 2022; https://openreview.net/forum?id=VmTYgjYloM), which proposed a principled framework to analyse and regularize the Fourier-sensitivity of models using the Fourier-decomposition of their input-Jacobian.

I kindly request the authors and editors to add discussion of this paper in the related work section for a more complete summary of the relevant literature.

---

### Decision · Action_Editor_Jhhv · 2023-12-05

**Recommendation:** Accept as is

**Comment:**

All three reviewers side with acceptance. The authors gave a thorough response to each reviewer, and the discussion resulted in a number of improvements, which have been included in the revision. The authors have summarized the changes with attribution to each review thread. In particular, the revision addressed issues with the claims and evidence to include citations, further comparisons, and content in the appendices.

The submission can be accepted essentially "as is" with a last round of trivial changes. This last round should adjust the formatting, to include the changes from the revision as normal text, and proofread the text to correct minor issues (for instance, the grammar glitch of "Exemplary, we show", the vocabulary of "sensibility" vs. "sensitivity", ...).

The AE thanks the authors and reviewers for engaging in the TMLR process to deliver an improved paper with agreement between its claims and evidence and a clear audience.

**Audience:**

There is an audience for this work comprised of those interested in robustness in general, adversarial training in particular, and signal processing. These topics are highlighted by the introduction, related work, and experimental results. While this work is related to prior studies of model filters and adversarial training, the particular method and findings here are informative w.r.t. existing paper.

**Claims And Evidence:**

All reviewers agree that the claims fit the evidence. Furthermore, the revision has satisfactorily resolved questions, added more experiments to relate filter frequency spectra with robustness alongside interpretability, and expanded on the motivation, discussion, and related work. The experiments cover several image classification models, datasets, and types of generalization with comparisons in the chosen scope of regularization methods for "native" robustness. Given the clear qualification of scope to regularization, without adversarial training or other types of perturbed or OOD data, the results indeed show convincing improvement. The relationship of the improvement to the regularization is confirmed by the analysis of filter and feature spectra. Limitations are noted and discussed within the conclusion of the paper and square with the claims.